# Fused-Linked and Spiro-Linked N-Containing Heterocycles

**DOI:** 10.3390/ijms26157435

**Published:** 2025-08-01

**Authors:** Mikhail Yu. Moskalik, Bagrat A. Shainyan

**Affiliations:** A. E. Favorsky Irkutsk Institute of Chemistry, Siberian Division of the Russian Academy of Sciences, 664033 Irkutsk, Russia

**Keywords:** fused heterocycles, spiro heterocycles, aziridines and azetidines, pyrrolidines, pyrazolines, imidazolines

## Abstract

Fused and spiro nitrogen-containing heterocycles play an important role as structural motifs in numerous biologically active natural products and pharmaceuticals. The review summarizes various approaches to the synthesis of three-, four-, five-, and six-membered fused and spiro heterocycles with one or two nitrogen atoms. The assembling of the titled compounds via cycloaddition, oxidative cyclization, intramolecular ring closure, and insertion of sextet intermediates—carbenes and nitrenes—is examined on a vast number of examples. Many of the reactions proceed with high regio-, stereo-, or diastereoselectivity and in excellent, up to quantitative, yield, which is of principal importance for the synthesis of chiral drug-like compounds. For most unusual and hardly predictable transformations, the mechanisms are given or referred to.

## 1. Introduction

Nitrogen-containing heterocyclic compounds are an important structural motif in a significant number of natural compounds and pharmaceuticals [1]. Most biologically active molecules (more than 80%) contain heterocyclic rings in their structure, with nitrogen-containing rings being the most common type [2]. This is due to their chemical stability, ability to be functionalized, and ability to form hydrogen bonds during biological processes [1,2,3,4,5,6]. More than half of all pharmaceuticals contain nitrogen-based heterocyclic structures [1,2,7,8]. This review discusses recent advances in the synthesis of fused- and spiro-linked N-heterocyclic systems (fused heterocycles, or having one common bond for any pair of adjacent rings, and spiro heterocycles, or having one common atom belonging to both rings), containing aziridine, azetidine, pyrrolidine, imidazoline, and pyrazoline heterocyclic motifs, and focuses on the academic publications that have been released in the last five years. The synthesis of compounds containing aziridine moieties has been extensively documented in scientific publications. These compounds possess great potential for the synthesis, and various methods have been developed over time to create molecules that incorporate these three-membered functional groups, which are spiro- or fused with four- to seven-membered heterocyclic rings. From a pharmaceutical standpoint, aziridines are of significant interest due to their promising biological activity [9,10]. Azetidines are a class of four-membered ring compounds that contain nitrogen and are widely used in medicinal chemistry [11,12]. They impart structural rigidity to the molecules, making them capable of unique chemical modifications. Incorporating rigid structures into drug design can lead to the development of more promising pharmaceuticals. While the incorporation of azetidine rings into complex molecules is desirable, there are limited and reliable methods available for synthesizing these highly strained rings [9,13,14]. Pyrrolidines are a group of nitrogen-containing cyclic compounds that have been extensively utilized in medicine to develop drugs for treating human illnesses. Pyrrolidines are of great importance due to their capacity to thoroughly explore the potential for drug development, which is made possible by the sp^3^ hybridization and 3D coverage provided by the non-planar structure of the pyrrolidine ring, a phenomenon known as “pseudorotation” [15]. A substantial number of these compounds have been identified through pharmaceutical research, and they demonstrate a high degree of specificity for various diseases. The presence of the pyrrolidine ring and its derivatives, such as pyrrolysine and pyrrolidinones, as well as imidazolines and pyrazolines [16,17], has been shown to contribute to this specificity [15,18]. Spiro- and fused-linked compounds are found in a variety of natural products and have a rigid spatial structure that makes them useful as ligands in asymmetric synthesis and catalysis [19]. However, there is no one-size-fits-all approach to the synthesis of these compounds, and selecting proper reactivity and compatibility of functional groups is a major challenge. From the viewpoint of stereoselectivity, the presence of a quaternary and often chiral spiro center is apparently the most critical concern [19]. Spiro- and fused-linked compounds that incorporate a fluorophore group and exhibit photoresponsive fluorescence are of particular significance as versatile detectors, innovative ligand systems, laser dyes, and electroluminescent devices. They also draw attention in the realm of macromolecular compounds [20]. This review aims to discuss the latest developments in the synthesis of a diverse range of fused and spiro-heterocyclic systems.

## 2. Fused- and Spiro-Linked 3-, 4-, and 5-Membered Nitrogen Heterocycles

### 2.1. Fused Aziridines and Diaziridines

Two principal routes to construct aziridines, including fused aziridines **3**, are nitrene **2** insertion into the C=C bond of alkenes **1** or insertion of carbene (generated, e.g., by dediazotation of diazoalkanes **5**) into the C=N bond of imines **4** (Figure 1). Both reactions are [2 + 1]-cycloaddition processes (Figure 1).

Another route to polycyclic fused aziridines containing the 1-azabicyclo[4.1.0]hept-3-ene motif (**8** or **10**) with a bridgehead nitrogen atom is an intermolecular aza-Diels–Alder reaction of 1,3-dienes **6** with 2*H*-azirines **7**. The reaction may be either intermolecular (Figure 2a) [21,22] or intramolecular (Figure 2b) [23].

An interesting Rh-catalyzed [4 + 1 + 1] sequential annulation was proposed recently for assembling highly fused aziridines **13** in up to 89% yield (Figure 3) [24].

A one-pot sequential transformation via decarboxylative Mannich reaction and oxidative C−H amination was shown to afford the products with fused aziridine and 1,2,3-oxathiazinane rings **17** with excellent diastereoselectivity (Figure 4) [25].

Similarly proceeds the reaction of benzo[*e*][1,2,3]oxathiazine 2,2-dioxides **14** in Figure 4 with α-azidostyrenes **18,** leading, after denitrogenation and hydrolysis of the intermediate iminium Cu(II) complex, to **17** (Figure 5) [26]. The same product is formed by the reaction of benzo[*e*][1,2,3]oxathiazine 2,2-dioxide with the sulfur ylide formed in situ by the action of a base on its precursor, salt [PhC(O)CH_2_SR_2_]^+^ Br^−^ [27]. The synthetic utility of this chemistry was demonstrated by gram-scale operation and further product derivatizations. Compounds can be used as an actin polymerization inducer [27].

A related process of phosphonium salt-catalyzed annulation of α-halogenated ketones and cyclic *N*-sulfonyl ketimines (saccharine derivatives) **19** via the C=N bond led to the products **21** of the *Mannich* reaction under mild conditions in up to quantitative yield and enantioselectivity [28,29]. Another group of researchers has shown that the structure of the products **22** of condensation of cyclic *N*-sulfonyl ketimines (saccharine derivatives) **19** with acetophenones **20** depends on the conditions, as shown in Figure 6 [30].

Fused aziridines are also formed by [2 + 1] cycloaddition of sulfur ylides RC(O)–CH=SR_2_ to saccharin-derived ketimines [31]; the reaction proceeds diastereoselectively in up to 94% yield. The one-pot three-component reaction of isatines **23**, α-amino acids **24**, and 2*H*-azirines **25** results in the formation of 1,3-diazaspiro[bicyclo[3.1.0]hexane]oxindoles **26** in high yields. The reaction occurs under mild conditions, tolerates a wide range of substrates, is regio- and diastereoselective, proceeds via the intermediate azomethine ylides generated in situ, and allows the construction of an unprecedented framework **26** (Figure 7) [32].

Using the procedure proposed earlier by the authors of [33], (1*R*,5*S*)-6-azabicyclo [3.2.0]hept-3-en-7-one **27** (a fused β-lactam, see next section) was converted into fused N-tosyl aziridine **29** via the ring opening of intermediate **28** and cyclization under the action of Chloramine-T (Figure 8) [33].

First, the product **29** of (1*R*,2*R*,3*R*,5*S*) configuration was formed, which was further converted to the *trans*-derivative of (1*R*,2*R*,3*S*,5*S*) configuration. Similar transformations were demonstrated for the isomeric nonfused β-lactam, 2-azabicyclo[2.2.1]hept-5-en-3-one [33]. Later, the same group of authors summarized in review [34] the results of the investigation of fused oxiranes and aziridines, focusing on regio and enantioselective ring-opening synthetic techniques for these compounds. Various N-containing polyheterocycles **31**, containing a fused aziridine motif, can be synthesized from simple pyrroles **30** by irradiation using fluorinated ethylene propylene flow reactor technology, as shown in Figure 9 [35]. Their miscellaneous transformations to compound **32**–**39** have also been considered.

The aziridine ring opening in 2-oxazolidinone-fused aziridines was used for the regio- and stereoselective synthesis of L-kijanosides, which are highly functionalized and hardly accessible natural deoxysugars [36]. The same group proposed the aziridine ring opening by the fluoride anion in 2-oxazolidinone-fused aziridines as a general approach to optically active, primary, secondary, and tertiary organofluorides with the skeleton of arabinose, which were precursors of various fluorinated amino acids [37]. Fused aziridines are capable of the ring opening. Thus, compounds **42** containing the aziridine ring fused with imidazole, pyrazine, or quinoxaline rings are shown to be able to give stable triplet biradicals upon irradiation in the solid state, which form azomethine ylides (Figure 10) [38].

1,2,3-Triazoles, readily available and stable, can be found in equilibrium with their ring-opened isomers, diazo compounds. These opened forms can be captured by various metal catalysts to produce corresponding metal carbenoid products by expelling nitrogen [39]. Rh-catalyzed aziridine ring expansion of aziridines fused with 1,3-oxazinan-2-one **43** and having the bridgehead nitrogen atom was shown to furnish dehydropiperazines **46** by the reaction with N-sulfonyl-1,2,3-triazoles **44** [40]. The reaction is highly diastereoselective, allowing us to overcome the problem of preparing stereopure piperazines as pharmaceutically important products. The ring expansion proceeds via a pseudo-1,4-sigmatropic rearrangement of an aziridinium ylide species **45**, (Figure 11). However, two years later, the same group re-examined their own results and showed that [3,9]-bicyclic aziridine formation competes with the supposed reaction course, and the final products are 9-membered heterocycles **47** with the aziridine and 3-(organsulfonyl)-6,7,8,9-tetrahydro-1,3,6-oxadiazonin-2(3*H*)-one rings [40], as shown in Figure 11, demonstrating how noncovalent interactions and restricted bond rotation in the aziridinium ylide intermediate **45** can unexpectedly change the reaction pathway.

Diaziridines have been known for more than 50 years, being first prepared by the reaction of cyclohexanone oxime O-sulfonic acid with ammonia. Here, we will mention only a few of the more recent publications; for earlier works, the reader can be referred to the reviews [41,42]. Thus, diamines of the norbornane or norbornene structure **48** react with *p*-chlorobenzaldehyde **49** in the presence of NBS as the oxidant to give a mixture of fused diaziridine **50** and pyrimidine products **51** in moderate overall yields, the former being the predominant one (Figure 12) [43]. The structure of the fused diaziridine **50** was proved by X-ray analysis.

The two nitrogen atoms in diaziridines in Figure 12 belong to the fused pyrrole and diaziridine heterocycles. Another type of fused diaziridine **53** can be prepared from cyclic secondary amines **52** and either arenesulfonamides in the presence of oxidant NBS or with bromamine-T and a catalytic amount of trifluoroacetic acid (Figure 13) [44]. Note that fused diaziridines of this type are hardly accessible by most of the existing methods.

Note that fused diaziridines **53** of the type shown in Figure 13 are hardly accessible by most of the existing methods. One of the methods that allow us to synthesize them was published very recently [45]. It is based on the reaction of homoallylic diazirines **54** with various radicals leading selectively to pyrrolines **57** via the addition to the C=C bond with subsequent ring expansion (intermediates **55**–**56**) or fused diaziridines **60** via the addition to the N=N bond (intermediates **58**–**59**) and hydrogen atom transfer (Figure 14).

### 2.2. Spiro Aziridines and Diaziridines

Spiro aziridines are represented in the literature mainly by the oxindole derivatives, which will be considered below separately. In 2017, a reaction of aziridination of cyclic enones **61** was proposed, affording new spiroaziridines **62**–**63** with a strained aziridine motif. The reaction is highly diastereoselective, scalable, proceeds under mild conditions, and tolerates a broad scope of substrates (Figure 15) [46]. A preliminary biological study of the products showed promising in vitro antibacterial activity against different pathogens.

The assembling of alkaloids possessing an unprecedented 1,5-diazaspiro[2.4]heptane fragment **67** with a spiro NH aziridine moiety and a 7-vinyl group by the thermal reaction of vinyl azides with tethered alkenes was reported [47]. Vinyl azides **64** are in situ converted to 2*H*-azirines **65,** which act as enophiles for intramolecular imino-ene addition to the C=C bond (Figure 16). The reaction is highly *cis*-stereoselective and stereospecific. However, in the course of further studies, it was found that the process is extremely sensitive to the substituents at nitrogen, which strongly affect the diastereoselectivity (*cis*/*trans* ratio (Figure 16)). This effect is most prominent for prenyl (3-methylbut-2-en-1-yl) vinyl azides, varying from 100/0 to 1.5/1 [48].

The 1,3-dipolar cycloaddition of ninhydrin **68** and α-amino acids **69** was shown to give spirocyclic heterocycles containing 3-azabicyclo[3.1.0]hexane and 2*H*-indene-1,3-dione motifs **70** [49]. The reaction proceeds stereoselectively under mild conditions with the formation of 3-azabicyclo[3.1.0]hexane-2,2′-indenes (at room temperature) (Figure 17). The antitumor activity of some products against the cervical carcinoma cell line was evaluated in vitro.

The authors of [50] disclosed an efficient diastereoselective synthesis of N-alkyl spiroaziridines by addition of primary amines to α,β-unsaturated ketones in the oxidative system I_2_/^t^BuOOH, similar to the reaction in Figure 15. Strange enough (maybe out of ignorance), the authors stated that ‘synthesis of spiroaziridines **74** has not been well explored so far’. N-Tosylimines TsN=CHAr **72** react with bicyclo[1.1.0]butyl sulfoxide **71** lithiated in situ to afford an intermediate **73,** which is cross-coupled with an aryl triflate through C-C σ-bond aminopalladation with concomitant aziridine **74** formation (Figure 18) [51].

In the last decade, a group of Indian chemists has published a whole series of papers on oxindoles with spiro-connected aziridine and oxindole rings [52,53,54,55,56,57,58,59,60,61]. Thus, based on the aza-Corey−Chaykovsky reaction of isatin-derived N-*tert*-butanesulfinyl ketimines **75** with sulfur ylides **76**, they elaborated a general strategy for the synthesis of chiral spiro-aziridine oxindoles **77** with excellent selectivity (*dr* ≥ 98:2), also applicable to the synthesis of chiral 3-substituted spiro-aziridine oxindoles with high, up to 98:2, (2*S*,3*S*) over (2*S*,3*R*) selectivity (Figure 19) [52]. The ^t^BuS(O) protecting group can be easily removed under mild conditions to afford product **78** (reaction *a*).

The N-sulfinyl azirine **77** in Figure 19 was oxidized to the corresponding N-sulfonyl azirine **79** with *m*-CPBA (*m*-chloroperbenzoic acid) and introduced in the reaction with indole to give, after the spiro-aziridine ring opening, unsymmetrical 3,3′-bis-indoles **80** (reaction *b*) [53]. The aziridine ring can be opened also by tetrabutylammonium fluoride to produce product **81** (reaction *c*) [54]. Other reactions of spiro-aziridine ring opening with various N-, O-, and S-nucleophiles as well as further transformations were studied [52,53,54,55,56,57,58,59,60,61]. The aziridine ring **79** in the sulfonyl derivatives in Figure 19 can not only be opened by different reagents but also undergo ring expansion to the 3-pyrrolyl ring by the action of allylsilanes in the presence of Lewis or Brønsted acids or allylmagnesium bromide [61]. Compound **79** can also be converted to the product with spiro-joined oxindole and 2-iminothiazolidine rings **82** (Figure 20) [62], or [2,3-*b*]-fused pyrrolyl rings [63,64]. A review summarizing some results of the last decade on epoxidation and aziridination of oxindole derivatives was published in 2020 [65].

As an alternative to aza-Corey−Chaykovsky reaction with ketimines in Figure 19, the Corey–Chaykovsky reaction of epoxidation of isatins was proposed, followed by the action of NH_4_OH and sulfonyl chlorides on the formed epoxide and, after the treatment with a base, affording the same N-sulfonyl derivatives of the spiro-aziridinated oxindoles as in Figure 19 [66]. Concluding this section, three works should be mentioned in which spirooxindole 2*H*-azirines **85**, rather than aziridines, were synthesized. The interest in 2*H*-azirines is due to their high ring strain and, hence, high reactivity as both nucleophiles and electrophiles, and in view of their presence in numerous natural compounds. In an earlier work [67], the Neber reaction was used to synthesize the target compounds. Later on, a modification was employed, affording the same spirooxindole 2*H*-azirines **85** ([68] and references therein) (Figure 21). Note that the work [67], in which (DHQD)_2_PHAL, hydroquinidine 1,4-phthalazinediyl diether, was used as a chiral catalyst, was the first enantioselective Neber reaction of O-sulfonyl ketoxime **84** and allowed the synthesis of spirocyclic oxindoles with the azirine motif **86** in good to excellent yields and with up to a 92:8 enantiomeric ratio.

### 2.3. Fused Azetidines and Diazetidines

Although fused azetidines are important heterocycles appearing in many antibiotics (penicillin, ampicillin, gelsemoxonine, calydaphninone, etc.), they are far from being thoroughly investigated. As highly strained compounds, they are highly promising candidates for ring-opening and ring expansion reactions, and, as such, have been reviewed [69]. Principal methods for the synthesis of fused azetidines are depicted in Figure 22. They include four-membered ring closure promoted by elimination of an easily removable group X from the CH_2_X substituent (in **87**) vicinal to the nitrogen atom in cyclic precursors (**88**) (e.g., reaction *a*), or, vice versa, generation of carbene, e.g., by dediazotation, and ring closure by its insertion into the N–H or C–H bond (**90** from **89**) (reaction *b*). For the examples of these and other types of fused and spiro-joined azetidines in earlier works, the reader can be referred to review [70].

Another example of the azetidine ring closure is presented by the intramolecular Mitsunobu reaction (Figure 23) [71].

4-Allenyl β-lactams **93** (already containing the azetidine ring in the molecule) can be fused with a pyrroline or tetrahydrofurane ring **94** via Au-catalyzed cyclization (Figure 24), as described in a review [72]. Other types of cyclizations resulting in an azetidinone ring fused with different 5- and 7-membered O-heterocycles are also reported.

The Lewis bases-catalyzed cycloaddition of allenoates CH_2_=C=CH–CO_2_R **96** to cyclic ketimines **95** under mild conditions (r.t., toluene) was developed [73]. Remarkably, the reaction proceeds either as [2 + 2] or [3 + 2] cycloaddition, depending on the catalyst, as shown in Figure 25. With DABCO (1,4-diazabicyclo[2.2.2]octane) as the catalyst and PPh_3_ the corresponding sultam-fused azetidines **97** are exclusively formed, whereas the triarylphosphine-catalyzed reactions give only dihydropyrroles **98** in the regiospecific manner, with the ester group in the α-position to nitrogen.

The use of organocatalytic protocols for the synthesis of enantiopure fused azetidines is limited. An example of chiral N-heterocyclic carbene (NHC)-catalyzed assembling of fused azetidines **102** and **106** is shown in Figure 26a [74]. Very recently, a chiral phosphoric acid (CPA)-catalyzed three-component reaction of anilines **103**, aldehydes **104**, and β-lactams **105** was reported (Figure 26b) [75].

Fused azetidinones **110** with a nitrogen atom belonging to both rings can be designed by insertion of a carbene at the β-position to nitrogen into the C–H bond at the β’-position to the same nitrogen atom, as shown, for example, in Figure 27 [76].

A large number of fused azetidines of the same type **113,** have been prepared via intermolecular aza Paternò–Büchi reaction by photoinduced Ir-catalyzed [2 + 2]-cycloaddition of the excited alkenes **111** to imines **112**, affording the **113** with the endocyclic nitrogen atom (Figure 28) [77] (see also [78]).

Another bicyclic heterocycle **116**, with fused N-protected azetidine isoxazole rings, substituted 7-Boc-2-oxa-3,7-diazabicyclo[3.2.0]hept-3-enes, was synthesized by [3 + 2]-cycloaddition of N-Boc azetidines **114** with imidoyl chlorides **115** in up to 91% yield and *dr* > 97:3 (Figure 29) [79] (Boc = *tert*-butoxycarbonyl protecting group, *^t^*BuOC(O)−).

The visible-light-induced dearomatization of indole-tethered O-methyl oximes **117** proceeds via triplet diradicals, which undergo intramolecular cyclization followed by [2 + 2] cycloaddition to give heavily condensed indoline-fused azetidines **121** as the kinetically controlled products shown in Figure 30, in up to quantitative yield [80]. Alternatively, 1,5-hydrogen atom transfer can occur to the thermodynamically controlled products.

1,2-Fused indoloazetidines **124** were obtained by the Rh-catalyzed cyclization of 1-azido-2-[(cyclopropylidene)methyl]benzenes **122** (Figure 31). The mechanism includes dediazotation with the formation of nitrene **123**, which is intramolecularly inserted into the C_sp2_–C_sp3_ bond of the cyclopropylidene moiety [81].

A large number of aziridine- and azetidine-fused bicyclic iminosugars were reported in the last decade and are described in the review [82]. A regiodivergent approach to fused 2- and 3-alkylideneazetines (**127** and **128**) was designed using the reaction of cycloaddition of intermediate 2- or 3-vinylazetines (**126**) to maleimide. The products of the elaborated three-step sequence (α-lithation, electrophilic addition, and [4 + 2] cycloaddition to maleimide) are formed in good yields and regio- and stereoselectivities (Figure 32) [83].

Not only fused azetidines but also fused 1- and 2-azetines are known, that is, four-membered N-heterocycles with the nitrogen atom only in the azetine motif or belonging to both fused rings, and with the C=C or C=N bond in the azetine ring. The works in this field published since 2018 have been summarized in a review [13]. Note that non-fused fluorinated 2-azetines **131** undergo original [2 + 2] photodimerization, resulting in the formation of bis-fused azetidines **132**–**133** in good to excellent yields (Figure 33) [84].

### 2.4. Spiro-Fused Azetidines

In the review [85], applications of pyrrolidine- and fused-pyrrolidine mimetics of piperazine and homopiperazine, including a number of fused azetidine derivatives, are discussed. A large series of spiro[3.3]heptanes containing an azetidine ring spiro-fused with cyclobutane, oxetane, thietane, or another azetidine ring (azaspiro[3.3]heptanes) with a nitrogen atom in either position was synthesized as potentially useful synthetic building blocks for drug design [86,87], and their molecular and conformational structure was discussed. The review [88] provides the reader with an up-to-date overview of the application of small rings, in particular, spiro and fused three- and four-membered rings, including azetidines, in medicinal chemistry. In the review [89], a comprehensive description of the biosyntheses of the azetidine-containing natural products, including those with fused and spiro-fused azetidine moieties, is described. A few examples of 2,5-diazaspiro[3.4]octanes with N-Boc-protected azetidine nitrogen were obtained by condensation of N-Boc-azetidine-3-one with *N*-benzylbut-3-yn-1-amine elaborated for a streamlined synthesis of C(sp^3^)-rich N-heterospirocycles via visible-light-mediated Ir-photocatalyzed reactions [90]. A new spirocyclization reaction for the synthesis of azetidine spirocycles was developed [91] and called ‘an elegant synthetic avenue’ to these compounds (**135**, **137**) (Figure 34) [92]. An azabicyclo[2.1.1]hexane intermediate formed as a single diastereomer, is converted to the final product. The studied reactions reveal the potential of the strain-release-driven spirocyclization strategy for rapidly assembling complex sp^3^-rich scaffolds.

The strategy using the inherent strain energy of a cyclic fragment [93,94] was applied to the synthesis of azetidine-containing spirocycles [91,95,96] based on transformations of the strained heterocycle azabicyclo-[1.1.0]butane, which is known for more than half a century but is experiencing a renaissance in the last decade [97,98]. Cyclic β-keto phosphonates **138** react with N-nosyl-O-(2-bromoethyl)hydroxylamine **139**, generated in situ from formaldehyde and nosylamide, affording 1,3-aminoalcohols **140**, which are converted into the spirocyclic **141** and bispirocyclic azetidines **142** via the Mitsunobu reaction (Figure 35) [99,100].

Much fewer works are known in which the products contain a fused diazetidine fragment, that is, a four-membered ring with two nitrogen atoms in different positions with respect to each other and to the fused rings. The review of 2019 [101] summarizes synthetic studies of 1,2-diazetidines and 1,2-diazetines (including fused and spiro-fused ones) since 1980. For example, 1,2-dicarbalkoxy-3-alkylidene diazetidines **143** enter [2 + 2] cycloaddition with tetracyanoethylene **144** to give spiro-fused diazetidines **146** (Figure 36) [102].

A large series of fused 1,2-diazetidines **149** with one nitrogen atom common for the two fused rings was obtained by the [3 + 1] cycloaddition reaction of (3,4-dihydroisoquinolin-2-ium-2-yl)amides **148** with aryl isocyanides **147** (Figure 37) [103].

The reaction of N-methyl-1,2,4-triazoline-3,5-dione **150** with acenaphthylene **151** expectedly leads to the formation of [2 + 2] diazetidine **154**, as shown in Figure 38 [85]. However, unexpectedly, the 2:1 adducts **156** and **157,** also containing the diazetidine motif, are formed, presumably, by trapping a relatively stable intermediate biradical **155** in Figure 38 by the initial reagent, N-methyl-1,2,4-triazoline-3,5-dione **150** [104]. A similar diazetidine is formed by the reaction of indene with N-phenyl-1,2,4-triazoline-3,5-dione.

Two Pd^0^-catalyzed C(sp^3^)-H reactions proceeding via [105] or resulting in the formation of [106] fused azetidines are presented in Figure 39. The former reaction leads to benzazetidines **159** as unstable intermediates that rearrange to benzoxazines **160** through 4π electrocyclic ring-opening and 6π electrocyclization. The second one proceeds as 1,4-Pd migration, followed by intramolecular Heck coupling, allowing to obtain, among a variety of substituted bicyclo[4,2,0]octene-fused azetidines **162**. The mechanistic studies point to a rate-limiting C(sp^3^)−H activation step.

A Pd(II)-catalyzed γ-C–H amination of cyclic alkyl amines **163** by oxidative addition/reductive elimination was reported to result in the diastereoselective formation of enantiopure highly fused azetidines **165** [107]. The reaction tolerates a range of functional groups. The mechanism involves an intermediate octahedral aminoalkyl Pd(IV) complex. Nucleophilic attack of the tosylate at the carbon atom bearing the Pd(IV) group forms the C–OTs bond, which in turn is displaced by the proximal amino group to form the final azetidine (Figure 40).

A large series of spiro-, fused-, and spiro-fused-azetidines **168** was synthesized by copper-catalyzed reaction of photocycloadditions of non-conjugated imines **166** and alkenes **167** [108]. The excitation occurs via metal-to-ligand charge-transfer to achieve [2 + 2] cycloaddition by selective alkene activation (Figure 41).

### 2.5. Fused 5-Membered Heterocycles

Cyclopropanes fused with lactam and pyrrolidine moieties are important pharmacophoric units in various pharmaceuticals, including ciproximide, boceprevir, amitifadine, and trovafloxacin [109]. Recently, authors of work [109] have developed a biocatalytic method using iron biocomplexes, such as myoglobin, for the asymmetric synthesis of these compounds. The method involves the use of mutational landscape analysis and iterative site-saturation mutagenesis of sperm whale myoglobin to create a cyclization reaction of allyldiazoacetamide derivatives **169** into the corresponding bicyclic lactams **170** with high yields and enantioselectivities up to 99% [109] (Figure 42):

The reaction covers a wide range of substrates. Substrates containing methyl, methoxy, and ethyl groups at the nitrogen atom yield bicyclic products with high or quantitative yields (90–99%) and excellent enantioselectivity (>99% *ee*) [109]. In this case, 3-azabicyclo[3.1.0]hexan-2-one **170** is formed [109]. The formation of 2-azabicyclo[3.1.0]hexan-3-one **170** was demonstrated in the presence of Pd(OAc)_2_ via intramolecular asymmetric hydrocyclopropanylation of the corresponding alkynes [110].

Allyldiazoacetamides **169** with an unprotected secondary amide group are difficult substrates for transition metal-catalyzed cyclopropanation reactions due to the catalyst poisoning via coordination with the metal and competition with carbene insertion into the amide N-H bond. Under these conditions, the reaction yields 31% of the product with more than 99% enantioselectivity. Diazoacetamides **169** containing non-activated olefin groups and aryl substituents yield products in 23–99% yields but with high enantioselectivity (90–99%). It is worth noting that substrates with substituents in the para, meta, or ortho-positions to the nitrogen atom of the phenyl group yield cyclopropyl-γ-lactams **170** with good to excellent yields (71–99%) and good to excellent enantioselectivity (90–99% *ee*) [109].

These biocatalytic transformations have also been realized in whole cells, allowing the implementation of enzymatic cyclization to form chiral cyclopropane-γ-lactams **170** and β-cyclopropylamines **170**, as well as cyclopropane-fused pyrrolidines **170**. These compounds are valuable building blocks and synthons in medicinal chemistry and natural product synthesis [109].

The reaction for the formation of condensed pyrrolidine β-lactones **173** in the presence of NHC is known [111] (Figure 43):

Nitrogen-containing bicyclic β-lactone products **173** have been obtained in good yields and excellent stereoselectivity. The reaction is an efficient method for synthesizing target structures [111].

The main step of the reaction is the addition of a nitrogen nucleophile **174** to an acylazolium cation **175**, which is catalytically generated by NHC [111] (Figure 44):

Both β-aryl and β-alkyl enals can participate in the reaction, yielding products **173** with acceptable yields and good enantiomeric ratios. The use of the products **173** for obtaining pharmacologically active derivatives can be achieved under mild conditions [111].

Recently, a method for the synthesis of a new polycyclic system **178** was demonstrated based on the 1,3-dipolar cycloaddition of unstabilized N-methylazomethyne ylide **177** with 2-R-3,5-dinitropyridine **176**. These compounds can be considered as potential nitric oxide donors having other types of biological activity due to their pyrrolidine and tetrahydropyridine fragments [112,113] (Figure 45):

The products were obtained through reactions with unstabilized azomethine ylides **177**, which were synthesized in situ from *N*-methylglycine and paraformaldehyde [112,113]. Compounds **178** with heterocyclic aromatic moieties were isolated, specifically those with C=C–NO_2_. In contrast to 2-unsubstituted **176**, the C=N fragment in these compounds is not involved in the reaction with the azomethine ylide **177**, representing the first synthesis of **178**. The compounds **178** with a hydrogenated pyrrolo[3,4-c]pyridine core have high antibacterial, anticancer, and cognitive-promoting potential for biological activity [112].

Recently, the synthesis of bicyclic fused pyrrolidines **181** through [3 + 2]-cycloaddition of an unstabilized azomethine ylide **180** with endocyclic electron-deficient alkenes **179** has been demonstrated. The products **181** are important in medicinal chemistry, as they include sulfonyl, trifluoromethyl- and fluorine-substituted derivatives, and oxygen-containing five- and six-membered heterocycles, which play a significant role in drug development and agrochemistry [114] (Figure 46):

Under acidic conditions (using TFA and CH_2_Cl_2_), the ylide precursor **180** forms a protonated intermediate **182** that eliminates MeOTMS ether (TMS = trimethylsilyl) to give the corresponding azomethine ylide **184**. This azomethine **184** then reacts with the electron-deficient alkene **179** through a [3 + 2] cycloaddition reaction to form the bicyclic pyrrolidine. In the presence of LiF, the reaction proceeds via a concerted pathway. Elimination of trimethylsilyl fluoride (TMSF) and LiOMe from **183** gives the azomethine **184**, which reacts with the alkene **179** [114] (Figure 47).

Some examples of chlorins have been synthesized similarly by the reaction of 5-(4-methoxycarbonylphenyl)-10,15,20-tris(pentafluorophenyl)porphyrin with azomethine ylide obtained from sarcosine and paraformaldehyde, followed by the hydrolysis of the ester group to obtain chlorin functionalized with benzoic acid. The reactions make it possible to obtain a number of N-functionalized chlorins in which the heterocyclic rings contain a condensed pyrroline fragment. The compounds can be used as photosensitizers in photodynamic therapy (PDT) of cancer and photodynamic inactivation (PDI) of microorganisms [115,116].

The same group of researchers [117], using acetylenes **186** as a starting material, demonstrated several examples of [2 + 2] and [2 + 3] cycloadditions of unstabilized azomethine ylides **180** and silyl enol ethers **185**. They obtained derivatives **187** containing polysubstituted cyclobutyl fragments in good yields with excellent diastereoselectivity [117,118] (Figure 48):

The synthesis of azasteroids is an important task in organic chemistry. Recently, new isoxazole derivatives **189**–**190** have been synthesized, which can be used as substrates for producing dehydroepiandrosterone derivatives [119] (Figure 49):

The key stage in this process is a multistep cycloaddition reaction proceeding under high pressure between an enol ether, nitroalkene, and different types of dipolarophiles. The reactions are regioselective, forming a mixture of two diastereomers through azinate intermediates **188**. Most often, only one of these isomers is isolated in pure form. If the dipolarophile has electron acceptor groups (like CO_2_Me or CN), the yield of the product is higher than with the electron-rich alkenes. After the cycloaddition, the formed semi-products (azonites) **189**–**190** are reduced to produce the desired steroid products [119].

Recently [120], a new, atom-economic, and highly stereospecific synthetic method for alkylating arenes and heteroarenes using the Friedel-Crafts reaction without the use of metal catalysts has been demonstrated. The reaction takes place in the presence of an aminium radical cation salt (Magic Blue), which opens the aziridine ring **191** through an S_N_2-type reaction in activated aziridines, followed by further alkylation with arenes or heteroarenes to produce 2,2-diaryl ethylamines. The reaction has been found to be useful in the synthesis of fused pyrrolidines **192**–**193**. When activated aziridines **191** are reacted under these conditions with 1,3-dimethylindole or benzofuran, they undergo Domino Ring Opening Cyclization reaction (DROC), leading to the formation of various nitrogen-containing compounds with high potential for biological activity. Reaction of aziridines with 1,3-dimethylindole yields the corresponding **192**. The reaction with benzofuran yields the corresponding **193** as a mixture of diastereomers with the overall yield of 46% [120,121,122] (Figure 50).

The synthesis of fused pyrrolidines was also achieved through a copper-catalyzed [3 + 2] cycloaddition reaction between 2-aryl aziridines **194** and cyclic silyl dienol ethers **195**. It was found that the reaction proceeded at 60 °C in the presence of Cu(OTf)_2_ as a catalyst and K_2_CO_3_ as an additive in a mixture of DCM and EtOH. Under optimized conditions, the method proved to be a versatile approach to the synthesis of bicyclic pyrrolidines **196** [123,124] (Figure 51):

2-Aryl-substituted aziridines **194**, with both electron-donating and electron-withdrawing substituents on the benzene ring, showed good to excellent yields and high diastereoselectivity. The reaction was successfully scaled up to the gram-scale level.

The impact of substituents at the C2–C6 positions on the silyldienol **195** substrate also demonstrated the relative flexibility of the process, with hydroindolones **196** being obtained in yields ranging from 40% to 83% with excellent exo-selectivity [123].

Recently, a novel approach for the synthesis of stereochemically enriched pyrrolidine- and benzo-fused sultams **200** has been developed. The method involves coupling of the previously unknown (o-fluoroaryl)sulfonyl aziridines **197** with 2-hydroxymethylpyrrolidines **198** through a series of steps, including aziridine ring opening and intramolecular nucleophilic aromatic substitution [125] (Figure 52):

This approach enables the reaction of various amino alcohols **198** to afford products with high chemo- and regioselectivity. It was found that the concentration of the solvent, the duration of the reaction, and the temperature were crucial factors. The opening of the aziridine ring and the S*_N_*Ar reaction occur through sequential intra- and intermolecular pathways. Increasing the reaction time and temperature was found to increase the yield of the reaction. It is important to note that the intramolecular ring-opening of the aziridine occurs at relatively high concentrations, while the subsequent intramolecular S*_N_*Ar reaction requires lower concentrations. Additionally, it is worth mentioning that while the opening of the aziridine ring can occur at room temperature, the reaction took five days to complete, whereas the use of MW activation allowed the reaction to be completed in just 30 min. Attempts to improve the reaction outcome by testing bases, such as CsF, K_2_CO_3_, K_3_PO_4_, DBU (1,8-diazabicyclo[5.4.0]undec-7-ene), and NaH, revealed that Cs_2_CO_3_ was the most effective. The use of both (*R*)- and (*S*)-prolinol as the amino alcohol **198** allowed the synthesis of a series of 6,10,5-fused tricyclic systems **200** with chiral center [125]. Recent advances in MW-assisted synthesis of heterocycles are described in the review [126].

A novel approach to the assembling of CF_3_-substituted pyrrolidinedione-fused pyrrolidines **203** has been devised. The method involves a three-component, decarboxylative cycloaddition of unstabilized *N*-unsubstituted azomethine ylides with readily available trifluoroketones **202** [127] (Figure 53):

Under optimized conditions, a series of chemical reactions were conducted involving a variety of amino acids **201**, 2-trifluoroacetophenones **202,** and maleimides **203** with the aim of synthesizing a range of trifluoromethyl pyrrolidines **204**. These compounds featured diverse substituents, and the yields of the products **204** varied within 55–77% and *dr* = 3–7:1. During the formation of the trifluoromethylated pyrrolidine-fused bicycles **204**, it was established that azomethine ylides could adopt either a W- or a U-configuration. By utilizing structurally rigid maleimides **203**, cycloaddition reactions resulted in the formation of only two diastereomers, which occurred through suprafacial reaction between W- or U-ylides **205** and maleimides **203**. The ratio of the diastereomers **204** was determined by NOE-2DNMR [127] (Figure 54):

Benzo[b]thiophene 1,1-dioxides **207** also undergo asymmetric 1,3-dipolar cycloaddition with azomethine **206** ylides as dipolarophiles. This approach, in the presence of Cu(I)-based catalysts and bisphosphine ligands, allows for the stereoselective synthesis of chiral tricyclic pyrrolidines fused to **208**, with four stereogenic centers [128] (Figure 55):

The products were obtained in good to excellent yields (up to 99%), with excellent diastereoselectivity and enantioselectivity (up to >25:1 *dr* and 99% *ee*). Several imino esters **206** and glycine derivatives were tested in the reaction, with varying ether groups (Me, Et, Bz, ^t^Bu). The substituent at the carboxylic group of **206** did not significantly affect the reaction. The corresponding chiral pyrrolidine-fused derivatives **208** of benzo[*b*]thiophene 1,1-dioxide **207** were obtained in high yields (90–94%) with excellent enantioselectivity (98–99% *ee*) using imino-*^t^*Bu-ether **206**, resulting in a notable improvement in diastereoselectivity. For imino-*^t^*Bu-ethers **206** containing both electron-withdrawing and electron-donating or electron-neutral substituents on the phenyl ring, cyclization occurred in high yields (85–99%) with good to excellent diastereoselectivity (8:1 to >25:1) and excellent enantioselectivity (93–99% *ee*). Sterically hindered *o*-Cl, *o*-Br, *o*-Me, and 1-naphthyl-substituted iminoesters **206** also gave products, although the enantioselectivity was slightly lower. Cycloaddition also occurred with heteroaryl-substituted azomethine ylides **206** derived from 2-thienyl and 2-furyl derivatives **208** with 88% *ee* and 99%, respectively [128].

The synthesis of **212** in good yields through one-pot, three-component reactions with isatins **210,** α-amino acids **211**, and cyclopropenes **209** has been reported recently [129] (Figure 56):

The reaction of equivalent amounts of methyl 2,3-diphenylcycloprop-2-enecarboxylic acid **211**, N-methyl substituted isatin **210,** and a small excess (25%) of L-proline **211** was studied as a model process. The best results were obtained by refluxing the mixture of the reagents in methanol for 4 h, which ensured a high yield (85%) and good diastereoselectivity. When cyclopropenes **209** with methyl substituents on the double bond or tetrasubstituted cyclopropenes **209** were used, cycloadditions did not yield the corresponding products. The presence of a substituent on the nitrogen atom of isatine leads to the formation of one diastereomer with yields of 48–85%. It should be noted that not only N-substituted or unsubstituted α-amino acids **211** but also the dipeptide Gly-Gly were used as amine components for the generation of azomethyne ylides. The anticancer activity of some of the obtained compounds was tested against the human leukemia cell line K562 [129].

A series of chiral benzosulfamidate-fused pyrrolidines **215** was synthesized using the Mannich/aza-Michael cascade reaction involving cyclic N-sulfimines **214**. The method is based on the reaction of cis-δ-formyl-α,β-unsaturated ketones **213** with cyclic N-sulfimines **214**, with TMS ether used as a catalyst [130] (Figure 57):

The reaction proceeds most effectively from the point of view of stereocontrol at −40 °C. The use of various solvents showed that the reaction medium has a significant effect on the conversion and stereoselectivity of the reaction. The best results were obtained with EtOAc and Et_2_O. Overall, the reactions of all substrates in Et_2_O afforded the corresponding benzosulfamidate-fused pyrrolidines **215** in good yields (71–96%) with good to excellent diastereo- and enantioselectivity (10:1–30:1 *dr*, 82–93% *ee*). The nature of the substituents on the aromatic ring had only a minor effect on the yields and stereoselectivity, with both electron-donating and electron-withdrawing substituents giving good results. Meta-substituents generally resulted in a higher reactivity and enantioselectivity than para-substituents. The approach allows the preparation of pyrrolidine derivatives, including those of pharmaceutical value [130].

The involvement of 2- and 3-alkylideneazetines **217** in the reaction with N-substituted maleimides allows the formation of unsaturated condensed tricyclic structures **220** with a pyrrolidine fragment, with high yields and regio- and stereoselectivity, through [4 + 2] cycloaddition [83,131] (Figure 58):

The reaction proceeds through the α-lithiation of vinyl azetidines **217** in the presence of *s*-BuLi, accompanied by β-elimination. When excess *s*-BuLi is present, a main azetinyl lithium intermediate **218** is formed, which reacts with an appropriate electrophile, such as H_2_O or TMSCl. The resulting diene **219** undergoes a [4 + 2]-cycloaddition reaction with electron-deficient dienophiles, forming condensed products **220** with high stereocontrol (up to 97:3 dr). This reaction allows the creation of several stereocenters, and, using in situ generated trans-2-butenyl lithium **218** and N-phenylmaleimide as the starting reagent, the products containing four consecutive stereocenters can be formed with excellent diastereomeric ratios and yields up to 96%. Substrates with bulky substituents also yield the expected products with good yields and stereoselectivity [83].

The synthesis of fused pyrrolidines **223** through the reaction of Michael addition products (semi-products were synthesized from α,α-dicyanoolefins and β-nitrostyrenes) has been elaborated. The resulting polycyclic fused pyrrolidines **223**, with three adjacent stereocenters, were obtained in high yields and with excellent diastereo- and enantioselectivity. The C(CN)_2_ group in **221** was oxidized using potassium permanganate in a mixture of acetone and water as a solvent at room temperature. The ketone **222** was reduced with zinc in acetic acid, resulting in a single stereoisomer in moderate yield and good enantioselectivity [132] (Figure 59):

Pyrazoline derivatives exhibit a wide range of biological and pharmaceutical activities, including antitumor, antibacterial, antifungal, antiviral, and anticancer activity. These nitrogen-containing five-membered heterocyclic compounds are well-known and have important applications in medicine. Various methods have been developed for the synthesis of pyrazolines, including the MW-assisted method, which offers several advantages, such as increased reaction rates [133]. Conventional heating of the reaction mixture to the temperature of boiling of acetic acid gives significantly lower yields [134]. An example of this synthesis is the preparation of new fused pyrazolines **227** by condensation of **224** with substituted benzaldehydes **225** under MW irradiation. The reactions yield chalcones **226**, which further react with phenylhydrazine to form **227**. Several of the obtained pyrazoline derivatives **227** show significant antibacterial activity, making them promising candidates for future drug development [133,135] (Figure 60):

The reaction of N-arylnitrilimines **228** prepared from trifluoroacetonitrile with levoglucosenone **230** proceeds through a [3 + 2] cycloaddition process, resulting in the formation of the corresponding fused pyrazolines **231**. In contrast to the similar reaction with non-fluorinated analogs, such as C(Ph),N(Ph)-nitrilimines **228**, the reaction with fluorinated derivatives, C(CF_3_), N(Ar), leads to the formation of stable pyrazolines in a chemo- and stereoselective manner, to produce the exo-isomer as a single product resulting from the [3 + 2] cycloaddition process. In all cases, the only products were tricyclic pyrazolines **231**, with yields ranging from satisfactory to good (47–88%). It is worth noting that, unlike non-fluorinated analogs, there was no spontaneous oxidation to trifluoromethyl-substituted pyrazolines **231** [136,137] (Figure 61):

A similar 1,3-dipolar cycloaddition reaction of nitrilimines **232** derived from hydrazonoyl chlorides was investigated [138,139] (Figure 62):

The process occurs under microwave radiation in the presence of N-phenyl- and N-methylmaleimides, as well as norbornene with specific substituents. The products exhibit exceptional optical characteristics, including fluorescence, positive solvatochromism, unique behavior in protic solvents like alcohols and water, and environmental sensitivity. Furthermore, these compounds **234**–**235** possess valuable features, such as intense green solid-state emission, making them ideal candidates for the development and synthesis of novel solid-state organic light-emitting materials. It is worth noting that the simultaneous demonstration of high fluorescence both in solutions and in solid form is a rare occurrence for both pyrazolines and other classes of organic fluorophores [138].

The environmentally friendly synthesis of imidazolidines **238** is achieved through a step-by-step process that involves the stereospecific opening of a ring and the addition of a C-H bond using aziridines **237** and secondary cyclic amines **236** under the influence of visible light in the presence of indazoloquinoline photoredox catalysts. The resulting fused imidazolines **238** exhibit high enantiomeric purity [140] (Figure 63):

Fused chiral pyrazolines **240** have also been synthesized by the reactions of hydrazine hydrate and **239** with moderate to good diastereoselectivity (up to 9.2:1 dr) and excellent enantioselectivity (up to 99% *ee*). The products **240** are tricyclic systems containing isoindoline, pyrrolidine, and pyrazoline moieties [141] (Figure 64):

### 2.6. Spiro-5-Membered Heterocycles

The reaction of isatin azomethine ylides with maleimides under asymmetric conditions in the presence of Cinchona alkaloid-based squaramide organocatalyst was studied [142] (Figure 65):

The reaction allows for the efficient synthesis of chiral pyrrolidine-fused spirooxindoles **244** in good yields (up to 89%) with excellent diastereo- and enantioselectivity (up to >20:1 *dr*, >99% *ee*). The approach allows the enantioselective assembly of synthetically and pharmaceutically important pyrrolidine-fused spirooxindoles **244** containing a pyrrolidine-2,5-dione moiety that contains four adjacent stereogenic centers, including one quaternary chiral center. As shown in Figure 65, the reaction proceeds in the presence of an organocatalyst together with 4A MS in CH_2_Cl_2_. Acidic additives significantly affect the enantioselectivity. In addition, under the action of stearic acid, isatin **241** condenses with the amine **242** to form the corresponding imine. The imine is then transformed into a 1,3-dipole via a 1,2-proton shift. Then, under the influence of Cinchona alkaloid catalysts and through multiple hydrogen bonds, the dipole undergoes a [3 + 2] 1,3-dipolar cycloaddition reaction [142]. A similar 1,3-dipolar [3 + 2]-cycloaddition of N-2,2,2-trifluoroethylisatin ketimines and maleimides occurs in the presence of phase-transfer catalysts, allowing the synthesis of a large set of trifluoromethylspiro-fused [succinimide-pyrrolidineoxindoles] in good yields (69–96%) and with excellent diastereoselectivity (>99:1*dr* for most cases). Unlike the previous work, a pre-prepared CF_3_-containing imine is introduced into the reaction with the maleimide [143]. Similar spiro-pyrolidinones are the basis for the synthesis of derivatives exhibiting insecticidal activity [144].

Chiral spirocyclic pyrrolidines can be synthesized by the use of CuBF_4_ in an asymmetric 1,3-dipolar cycloaddition reaction between azomethine ylides and various heteroatom-containing azetidines with exocyclic alkenyl groups. By adjusting the ligand and maintaining the catalysis conditions, the *exo*- or *endo*-adducts can be obtained. Using a ligand with an oxazolidine moiety yields *exo*-products, while a pyrazoline ligand results in the endocyclic products. A wide range of functionalized spirocyclic pyrrolidine azetidines has been successfully synthesized with high yields (up to 99%) and excellent enantioselectivities (up to 99% *ee*) [145] (Figure 66):

The development of techniques for producing spirocyclic structures is crucial for creating analogs of natural biologically active substances and alkaloids. For instance, spirooxindole-pyrrolidine derivatives exhibit a broad spectrum of biological activity found in a diverse family of alkaloids and natural products. Among these compounds are Alstonisine, Horsfiline, Coerulescine, and Elacomine. For instance, spirotryptostatins A and B, isolated from Aspergillus fumigatus, completely halt the transition from the G2 phase to the M phase in mammalian tsFT210 cells [146]. Synthetic derivatives of these natural substances are often more potent and specific than their natural counterparts. For example, the spirooxindole-pyrrolidine derivative MI-77301, which inhibits murine double minute 2 (MDM2), is currently undergoing phase I clinical trials [146]. The synthesis of spirooxindole-pyrrolidines can be achieved through a one-pot process involving the reaction between aziridine and 3-ylideneoxindole **248**. This reaction is highly efficient, with yields reaching 95% and diastereoselectivity exceeding 20:1 (Figure 67):

The synthesis of spirooxindole is a single-step process that involves the thermolysis of aziridine **249** to generate a 1,3-dipole. 1,3-Dipole then undergoes a 1,3-dipolar cycloaddition with 3-ylideneoxindole **248** as the dipolarophile, resulting in the formation of the target spirocycle **250**. This approach not only expands the range of applications for aziridines **249** but also provides an alternative method for producing pharmacologically significant spirooxindole-pyrrolidines **250** [146]. Additionally, Lewis acids can be employed to activate aziridines **249** for the cycloaddition process. For instance, in the presence of Sc(OTf)_3_, a [3 + 2]-annulation reaction between certain *exo*-glycals and aziridines has been successfully performed. *Exo*-glycals derived from D-ribose, D-galactose, and uridine have also been demonstrated to produce spiroheterocycles containing a pyrrolidine moiety with high efficiency when the reaction is carried out at −20 °C in dry DCM [147]. Some variations in a similar process are described in [64]. A novel approach to synthesizing spiro-1,3-benzothiazinoxindoles involved a unique rearrangement of cyclic ketimines derived from saccharin (SDCI) and 3-chlorooxindoles [148] or oxindoles and nitroalkenes [149,150,151]. The use of oxindoles in organic synthesis is described in the reviews [152,153,154].

The reaction of aziridines **252** and alkynyl alcohols or amides **251** under mild conditions in the presence of Au(I) catalyst and Lewis acids was studied. The reaction produces spiro nitrogen-containing heterocycles **253** with high stereoselectivity [155] (Figure 68):

Upon optimizing the conditions, it was found that the reaction proceeded smoothly when the alkyne **251** was slowly added to the reaction mixture in the presence of Ph_3_PAuNTf_2_. Six different Lewis acids have been tested, showing Yb(OTf)_3_ to be most effective. It yielded the desired product in nearly quantitative yield as a single diastereomer. The effect of substituents in the starting aziridines **252** has also been studied. For substrates **251** with para-substituents, the reaction yielded the target products in yields ranging from 72% to 99%. Substituents such as fluorine, chlorine, and bromine, as well as electron-withdrawing groups such as NO_2_ and CF_3_ and electron-donating groups such as methyl and methoxy, did not affect the reaction yield. Ortho and meta substituents also did not affect the reaction, leading to excellent yields of the corresponding products. A substrate **251** containing a 2-naphthyl group was found to be suitable for this reaction. In this case, both 2,2-dimethyl ether- and N-methylsulfonyl-substituted aziridines **252** gave the products with high yields. Fluorinated alkynyl alcohols **251** could also produce the desired product **253** with an acceptable yield. All reactions proceeded in one hour without changing the diastereoselectivity [155].

Another example is a multistep assembly of spiro-pyrrolidines. The process involves the reaction of cyclic carboxylic aldehydes **264** (Mannich reaction). The reaction enables the synthesis of spirocyclic aminolactones **257**, which can be further transformed into pharmaceutically important spiro[4.6]cyclic 3-aminopyrrolidines **260**. 3-Aminopyrrolidines **260** serve as crucial building blocks for drug development and the synthesis of organocatalysts [156] (Figure 69):

The synthesis of the spirocyclic compound begins with the reaction between cycloheptanecarbaldehyde **254** and an imine **255** with equivalent amounts of the reactants. The reaction takes place in the presence of an organocatalyst and CF_3_CO_2_H (80 mol%). Subsequently, a reduction occurs in the presence of NaBH_4_, resulting in the formation of a lactone **257**. The resulting product is then placed in toluene and subjected to heating for 12 h in the presence of 2-hydroxypyridine, which acts as a bifunctional catalyst. The reduction in spirolactone with LiAlH_4_ leads to the formation of the diol **258** in 79% yield. Further treatment with equivalent amounts of mesyl chloride directly results in the formation of the aziridine mesylate **259**. After the replacement of the mesylate by benzylamine and the subsequent intramolecular aziridine ring opening, 3-aminopyrrolidine **260** was synthesized in 69% yield in two steps by performing the reaction in 2,2,2-trifluoroethanol without any additional base other than benzylamine [156]. The use of different catalysts in the synthesis of spiroheterocycles is described in the review [157].

Recently, a novel approach to the synthesis of β-spirocyclic pyrrolidines **264** from N-allylsulfonamides **261**, halogenating agents, and exocyclic olefins **263** has been unveiled [158]. The presence of a side halomethyl group in the product **264** allows a variety of chemical transformations to be performed. Moreover, as demonstrated in this study, the method can be employed to produce spirocyclic derivatives **264** of pharmaceuticals, including celecoxib, valdecoxib, and methazolamide derivatives. The synthesis is conducted in a flow reactor on a scale of tens of grams, utilizing high-power light-emitting diodes, minimizing the potential risks associated with handling N-halogen intermediates [158] (Figure 70):

The work [159] shows a method for synthesizing spirocyclic imidazoloquinolines **269** from readily available 2-(methylsulfanyl)imidazolones **268** with various substituents in the benzylidene fragment with good yields [159] (Figure 71):

The reaction proceeds as a [1,5]-hydride shift from the NMe of the **267** group initiated by coordination of Sc(OTf)_3_ with the imidazolone oxygen of the **271** atom and followed by cyclization. The products contain the SMe group, which allows modification of these compounds [159] (Figure 72).

The synthesis of similar compounds via cycloaddition of quinoline ylides with arylideneimidazole-4-ones is demonstrated in the work [160].

An efficient method for the construction of fluorovinylspiro[imidazole-indene] derivatives **274** in the presence of Rh(III) catalyst was proposed via C−H functionalization of 2*H*-imidazoles **272** with difluoromethylene alkynes **273** [161]. The reaction is a simple method for the formation of fluorine-containing substituents incorporated into a complex heterocyclic framework bearing several stereocenters (Figure 73):

The reaction occurs similarly with acetylenes **273** with nonfluorinated substituents [162].

In [163], the first example of a [3 + 2] cycloaddition reaction of donor/donor diazo derivatives **275** with alkenes was reported, resulting in a series of (spiro)pyrazolines **276** or **277** (Figure 74):

Methods for obtaining donor/donor diazo derivatives from N-tosylhydrazones **275** for introduction into [3 + 2] cycloaddition reactions, according to the authors of the paper [163], were not available before this study. The applicability limits of [3 + 2] cycloaddition reactions between various N-tosylhydrazones **275** and alkenes for the synthesis of (spiro)pyrazolines were shown. Both EWG-substituted and EDG-substituted N-tosylhydrazones readily react, providing functionalized spiropyrazolines. The reaction also involves heteroaromatic cyclic ketones 6,7-dihydro-4-benzothiophenone, BOC-protected 1,5,6,7-tetrahydro-4*H*-indol-4-one and 7,8-dihydroquinolin-5(6H)-one, 4-chromanone and 2-methyl-1-tetralone. N-tosylhydrazones **275** synthesized from 1-indanone, 1-acenaphthenone and benzocycloheptanone also give reaction products.

First, the N-tosylhydrazone **275** formed as a result of the condensation reaction between the ketone and 4-tosylhydrazide is introduced into the reaction mixture. After deprotonation, it gives the corresponding N-tosylhydrazone anion **278**. Under the action of DBU and visible light (456 nm), homolytic cleavage of the N–S bond occurs. Next, donor/donor diazo species **279** are formed in situ, which then undergo [3 + 2]-cycloaddition with the alkene to give the desired spiropyrazoline **277** [163] (Figure 75):

Some examples of the use of hydrazones and diazo compounds in the synthesis of spiro-pyrazolines are presented in the works [164,165,166].

The reaction of 1*H*-pyrrole-2,3-diones **281** with phenylurea has been studied, followed by cyclization in the presence of sodium methoxide, leading to the formation of a 1,3,6-triazaspiro[4.4]nonane **284** structure with various functional groups [167] (Figure 76):

Compound 281 reacts with phenylurea at a 1:1 molar ratio by reflux in dry toluene for 5–10 min (until the color of the substrate disappears). The reaction results in the addition of the primary amino group of phenylurea, forming **282**. Sodium methoxide is used in a two-fold excess, as one equivalent is consumed for the formation of enolate, while the remaining equivalent deprotonates the NH group, forming an amide anion **283**. The latter undergoes intramolecular cyclization, forming a new amide bond and closing the five-membered ring, resulting in spiro compounds **284** [167]. Pyrrazoline spirocyclic fragments are found in natural alkaloids [168] and drugs [169].

One of the interesting examples of the synthesis of fused-linked or spiro-linked products from the same reagents, depending on the conditions, is presented in the works [170,171].

The reaction of camphene **286** with triflamide **285** and further treatment of the formed bromoamidine with Cs_2_CO_3_ gave the product of rearrangement, solvent (MeCN) interception, and fused camphane and quinazoline rings **287** [170,171]. This was the first synthesis of the product with fused quinazoline and terpene fragments. Replacement of the base by Cs_2_CO_3_ and carrying out the reaction in one-pot fashion allowed us to obtain the fused quinazoline products in up to 85% yield (Figure 77).

An unexpected result was obtained when using a three-fold excess of camphene **286** and NBS with respect to sulfonamide. Spirocyclic unrearranged imidazoline products (major, up to 70%) and rearranged fused azetidine products (minor, up to 27%) were isolated (Figure 78) [170]:

The difference in the reaction course in Figure 77 and Figure 78 is due to the different structure of the cation formed by electrophilic bromination: in the former case, the bromine atom is coordinated to Cs^+^, and the carbocation suffers skeletal rearrangement. In the latter case, the rearrangement of the formed bromonium ion is a side process leading to a minor azetidine product due to the attack of sulfonamide on the rearranged cation [170].

## 3. Conclusions

To summarize, the synthesis of fused- and spiro-linked N-heterocyclic systems is a rather non-trivial and creative task. The reviews on the chemistry of heterocycles, published earlier [13,34,41,65,72,82,101,126,152,153,154,157], also address the issues of synthesis and properties of some fused and spiro-heterocyclic systems. However, these reviews focus on specific compound classes of substrates or methods.

The synthesis of aziridines is based on nitrene insertion into the C=C bond or insertion of carbene into the C=N bond. Decarboxylative Mannich reaction and oxidative C−H amination are used for the synthesis of polycyclic aziridines. One approach to the synthesis of spiro-aziridines is the use of methylidene derivatives. The intra- and intermolecular Mitsunobu reactions provide an approach to the fused and spirocyclic azetidines, respectively. Acetylenic compounds are much less studied than alkenes or azomethines for the synthesis of the small fused and spiro N-heterocycles. The main starting materials for the synthesis of condensed pyrrolidines include enals, α-amino ketones, azomethyne ylides, aziridines, oxindoles, imines, β-nitrostyrenes, and hydrazonoyl derivatives of pyrazolines. The synthesis of spiropyrrolidines relies on the use of aziridines, oxindoles, imines, and enamides as reactants. These compounds are found in various important biological substances, such as physostigmine (also known as ezerine), a cholinesterase inhibitor. Asenapine, used in the treatment of bipolar disorder and schizophrenia, and other antiviral medications that inhibit hepatitis C virus proteases, are also included. Additionally, spirooxindole pyrrolidines like Horsifilin and Coerulescine are used as analgesics and local anesthetics. This gives an impetus to develop the design of structurally interesting and potentially biologically active new compounds based on these substrates. Of great importance is the selectivity of the reactions. Ideally, the products should be formed not only in high yield but also with high diastereo- or/and enantioselectivity.

An important trend in the chemistry of 3- and 4-membered heterocycles in the near future will be the application of biosynthetic techniques. The technology of reaction activation by visible light (blue LED) is also largely used for the synthesis of the compounds under consideration. The strategy of harnessing the inherent strain energy within the cyclic fragment for the synthesis of heterocyclic elements based on transformations of small heterocycles remains relevant and is experiencing a revival. In recent years, there have been no new methods discovered for the synthesis of small heterocyclic compounds with two nitrogen atoms. This task remains relevant, as diazetidines and diazeridines could become important building blocks for introducing two nitrogen atoms into larger heterocyclic compounds.

Significant progress has been made in the chemistry of five-membered rings, as this fragment is commonly found in natural products and drugs. Therefore, much time has been dedicated to studying methods for constructing or incorporating a pyrrhodine ring in these compounds.

The main goal of the present review was not so much to show a huge variety of fused and spiro structures, which is clearly evident from the figures above, but rather to provide the reader with a guiding line for further synthetic studies in the field.

## Figures and Tables

**Figure 1 ijms-26-07435-f001:**
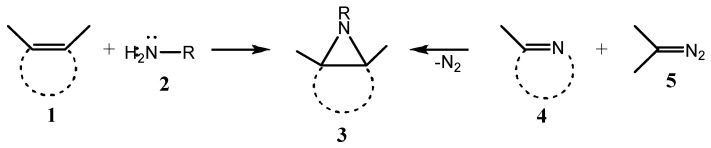
Two alternative routes to fused aziridines.

**Figure 2 ijms-26-07435-f002:**
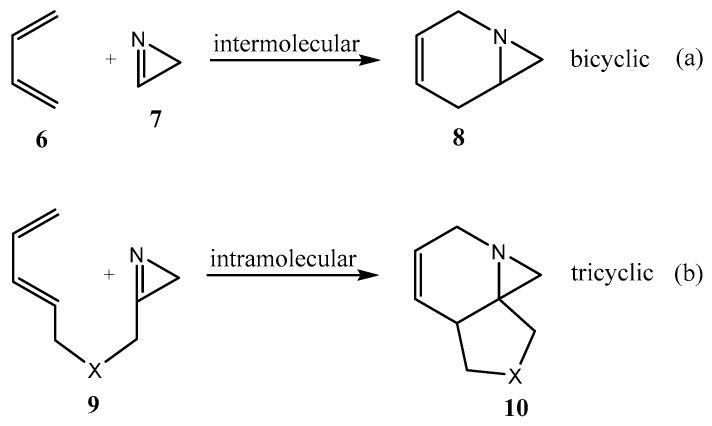
Inter- (**a**) or intramolecular (**b**) aza-Diels–Alder cycloaddition as a route to polycyclic ring systems with bridgehead nitrogen atoms.

**Figure 3 ijms-26-07435-f003:**
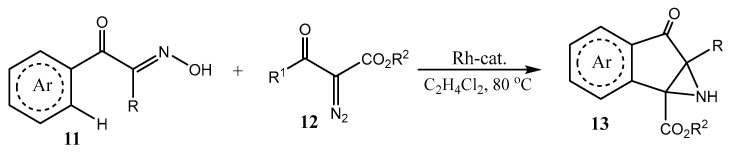
Rh-catalyzed successive cyclization with the formation of indano[1,2-*b*]azirines **13**.

**Figure 4 ijms-26-07435-f004:**
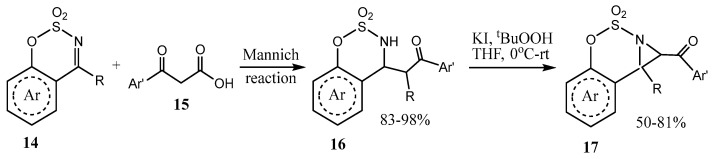
Reaction of cyclic imines with β-ketoacids and oxidation of the Mannich adduct to fused aziridines.

**Figure 5 ijms-26-07435-f005:**
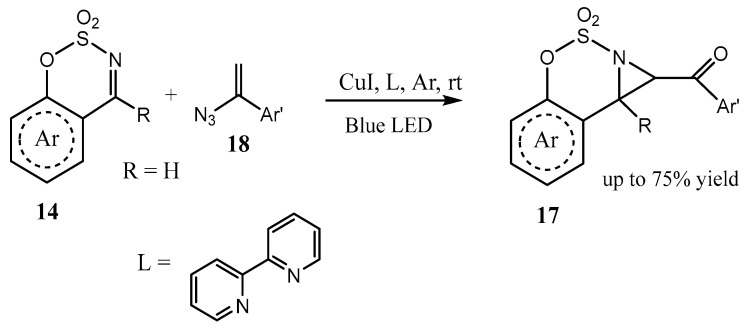
Cu(I)- and blue-LED light-catalyzed aziridination of cyclic N-sulfonylimines with vinyl azides into the sulfamidate-fused aziridines.

**Figure 6 ijms-26-07435-f006:**
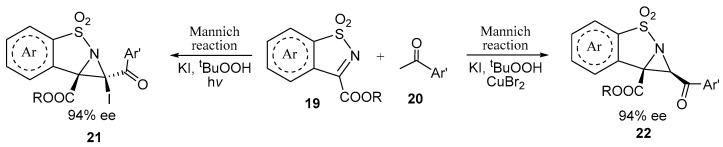
Condition-dependent diastereoselectivity of the reaction of saccharines with acetophenones.

**Figure 7 ijms-26-07435-f007:**
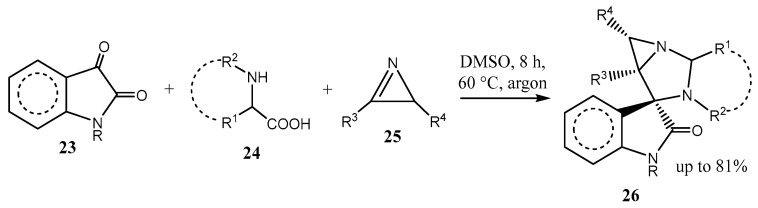
One-pot assembling of 1,3-diazaspiro[bicyclo[3.1.0]hexane]oxindoles.

**Figure 8 ijms-26-07435-f008:**
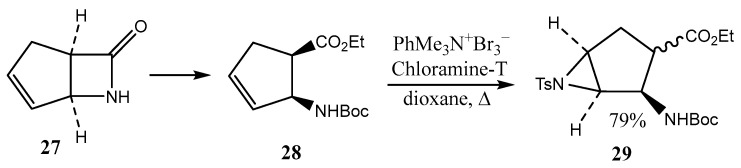
Aziridino amino ester from N-protected cyclopentene β-amino ester.

**Figure 9 ijms-26-07435-f009:**
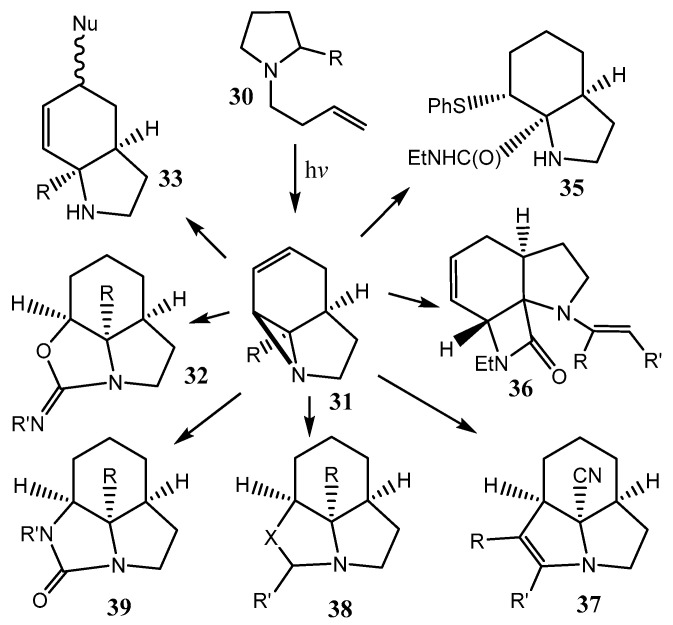
Ring-opening and cycloaddition reactions of highly strained fused aziridines.

**Figure 10 ijms-26-07435-f010:**
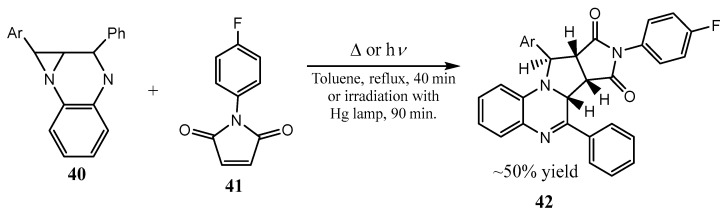
Thermal or UV-induced activation of stepwise transformation of fused aziridines.

**Figure 11 ijms-26-07435-f011:**
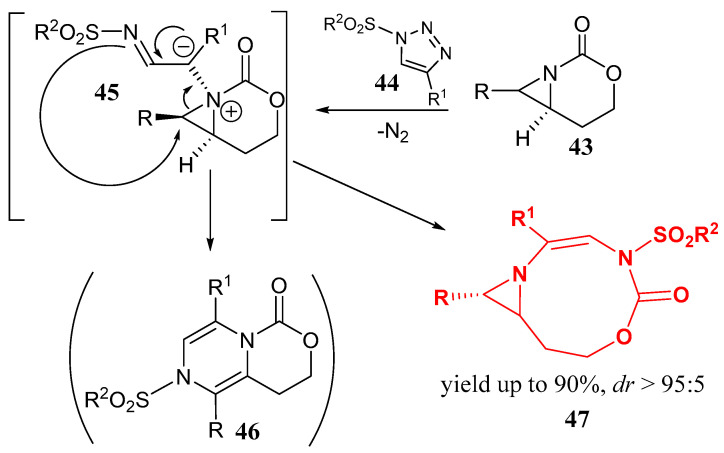
The originally supposed [19] and reexamined [20] 3-oxa-1-azabicyclo-[4.1.0]heptan-2-one to 6,7,8,9-tetrahydro-1,3,6-oxadiazonin-2(3*H*)-one ring expansion.

**Figure 12 ijms-26-07435-f012:**
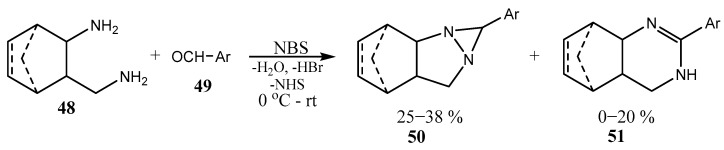
Diaziridine (major) **50** and pyrimidine (minor) **51** products of NBS-induced oxidative cyclization.

**Figure 13 ijms-26-07435-f013:**
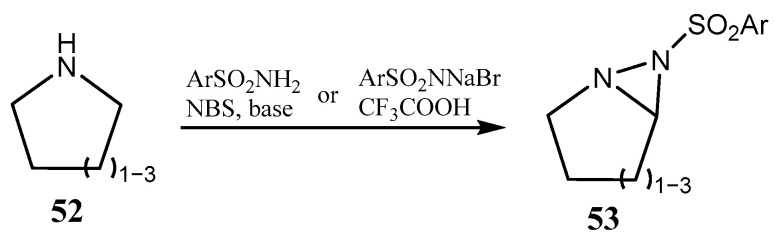
NBS-induced synthesis of fused diaziridines from cyclic amines and arenesulfonamide in the presence of base, or with bromamine-T and trifluoroacetic acid.

**Figure 14 ijms-26-07435-f014:**
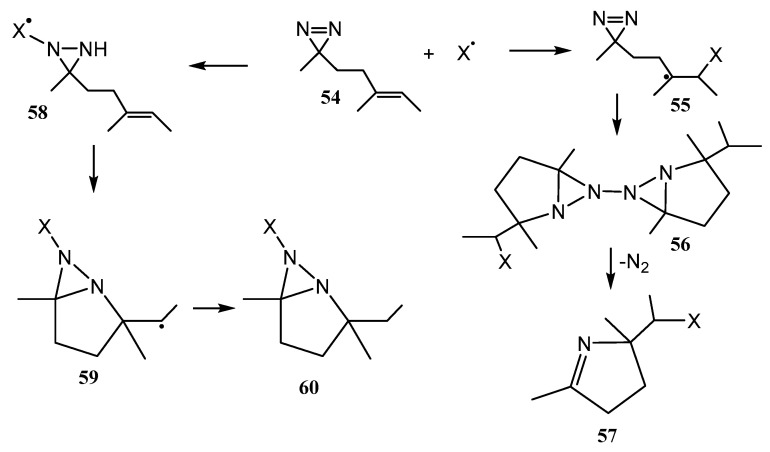
Diaziridines from homoallylic diazirines via the addition to the C=C bond and hydrogen atom transfer.

**Figure 15 ijms-26-07435-f015:**
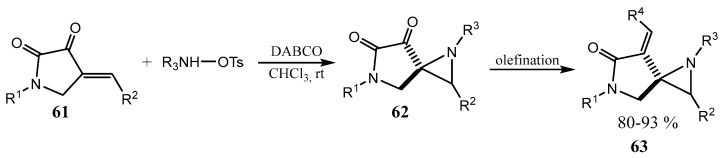
Highly diastereoselective aziridination of cyclic ketones.

**Figure 16 ijms-26-07435-f016:**
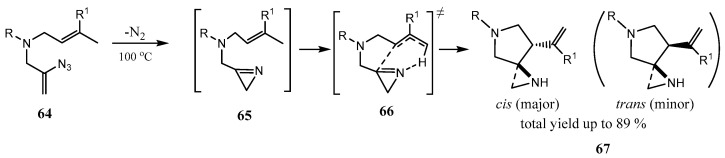
Formation and intramolecular cyclization of azirines **65**; up to 86% yield and the *cis*/*trans* ratio from 1.5:1 to 100:0.

**Figure 17 ijms-26-07435-f017:**
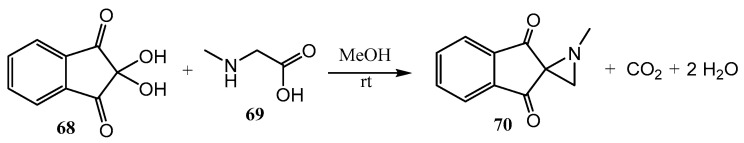
Spiroaziridines from ninhydrin and α-amino acids at room temperature versus azomethine ylide at reflux in methanol.

**Figure 18 ijms-26-07435-f018:**
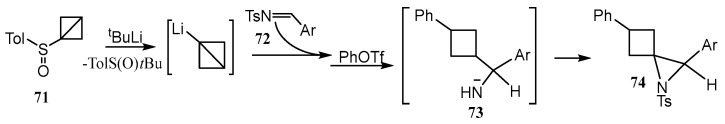
A strain-release-driven aziridination of folded bicyclobutane to spirocyclobutylaziridine.

**Figure 19 ijms-26-07435-f019:**
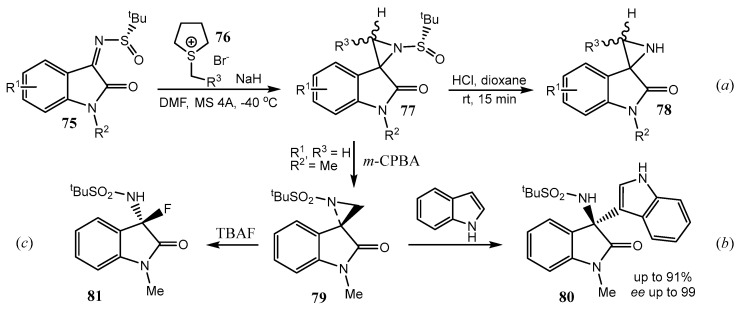
Synthesis and transformations of the isatin-based spiroaziridines. The ^t^BuS(O) protecting group removing (**a**); Oxydation N-sulfinyl group to N-sulfonyl group (**b**); Aziridine ring-openinig (**c**).

**Figure 20 ijms-26-07435-f020:**
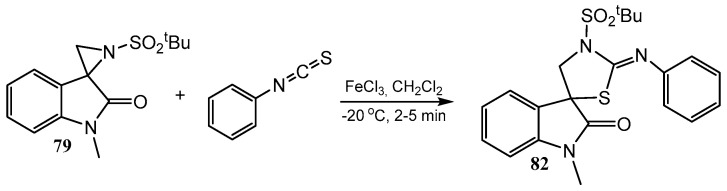
Spiro[indoline-3,5′-thiazolidin]-2-ones via aziridine ring-opening/ring-closure.

**Figure 21 ijms-26-07435-f021:**
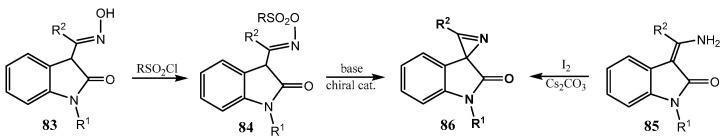
Two alternative approaches to the syntheses of spirooxindole 2*H*-azirines.

**Figure 22 ijms-26-07435-f022:**
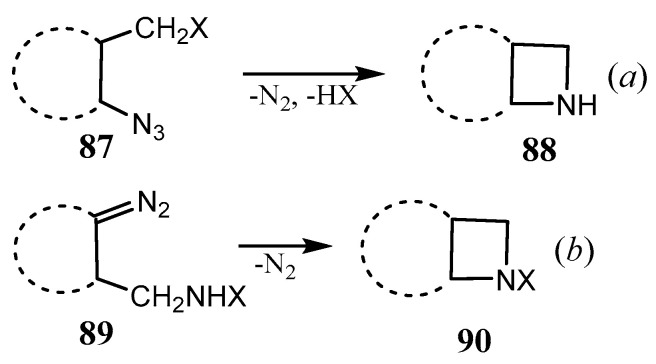
Principal approaches to fused azetidines **88** and **90** via ring closure and elimination of a leaving group X of **87** (**a**), or insertion of in situ generated carbene into N–H or C–H bond of **89** (**b**).

**Figure 23 ijms-26-07435-f023:**
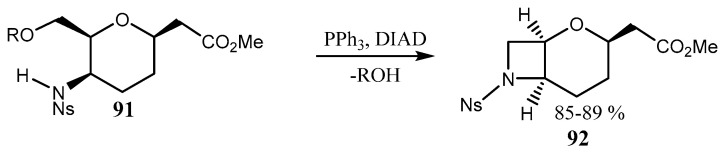
Fused azetidine **92** formation via intramolecular Mitsunobu reaction.

**Figure 24 ijms-26-07435-f024:**
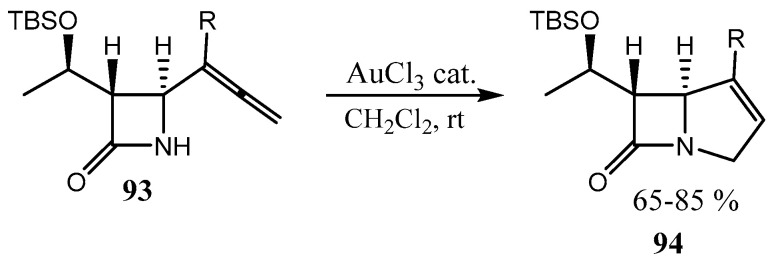
Gold-catalyzed cyclization of 4-allenyl-2-azetidinones into bicyclic β-lactams.

**Figure 25 ijms-26-07435-f025:**
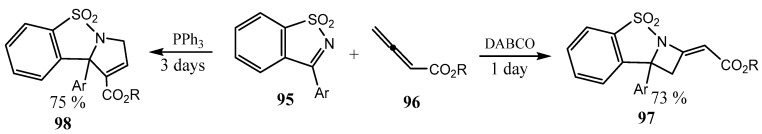
PPh_3_-catalyzed [3 + 2] cycloaddition or DABCO-catalyzed [2 + 2] cycloaddition of allenoates **96** to cyclic ketimines **95**.

**Figure 26 ijms-26-07435-f026:**
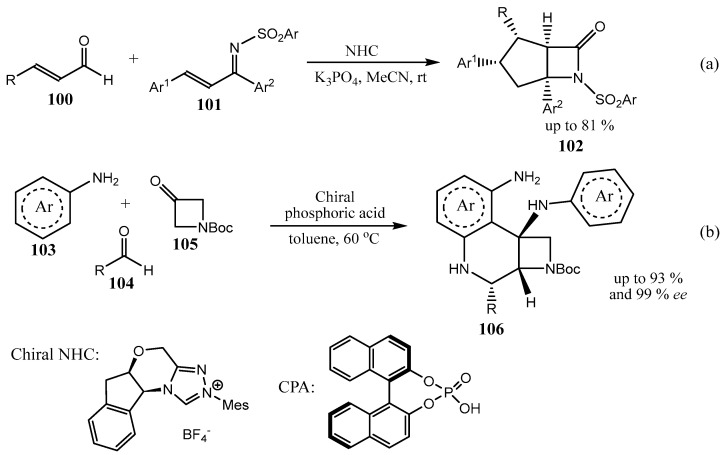
Synthesis of β-lactams fused with spirocyclopentane oxindoles (R) (**a**) and CPA-catalyzed multicomponent reaction of anilines, aldehydes, and azetidinones (**b**).

**Figure 27 ijms-26-07435-f027:**
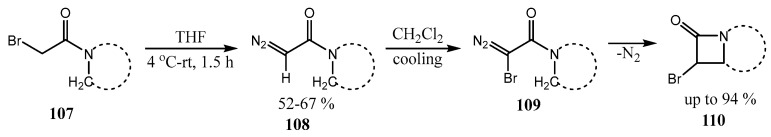
Synthesis of diazoacetamides, their bromination, and thermolysis to the fused azetidinones.

**Figure 28 ijms-26-07435-f028:**
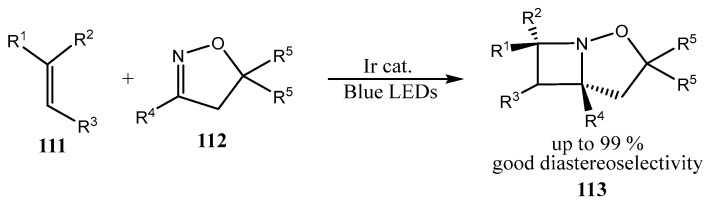
Ir-catalyzed photoinduced intermolecular aza Paternò–Büchi reaction.

**Figure 29 ijms-26-07435-f029:**
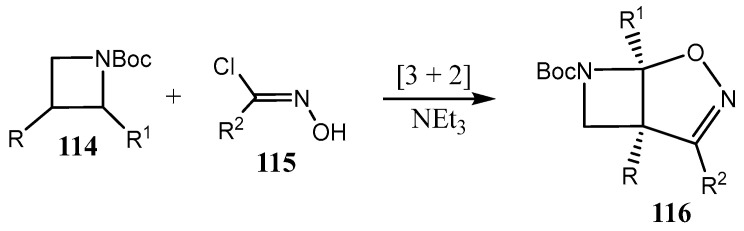
[3 + 2]-Cycloaddition of N-Boc azetidines **114** with *N*-hydroxynimidoyl chlorides **115**.

**Figure 30 ijms-26-07435-f030:**
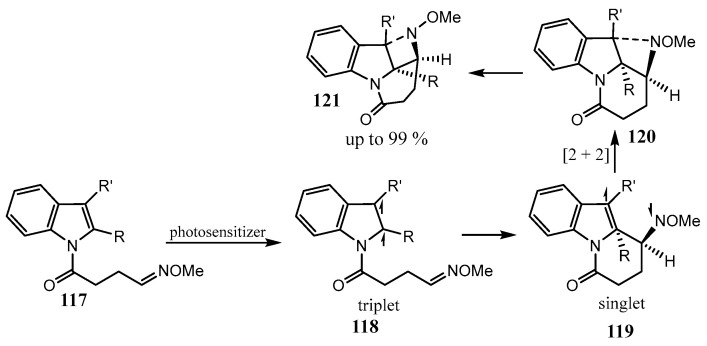
Blue LED-induced [2 + 2] cycloaddition reaction. CH_2_Cl_2_, rt, argon, 8 h.

**Figure 31 ijms-26-07435-f031:**
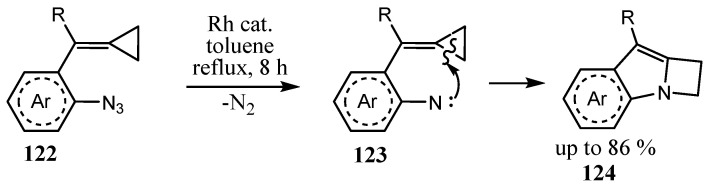
Rh-catalyzed cyclization of 1-azido-2-(cyclopropylidenemethyl)benzenes.

**Figure 32 ijms-26-07435-f032:**
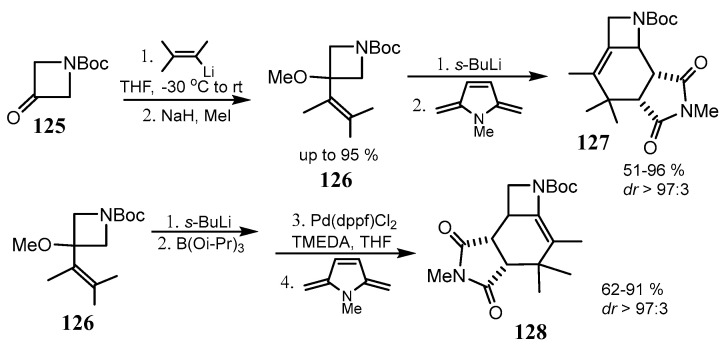
Synthesis of 3-vinylazetidine precursors **126** and fused 2-alkylideneazetidines **127**–**128**.

**Figure 33 ijms-26-07435-f033:**
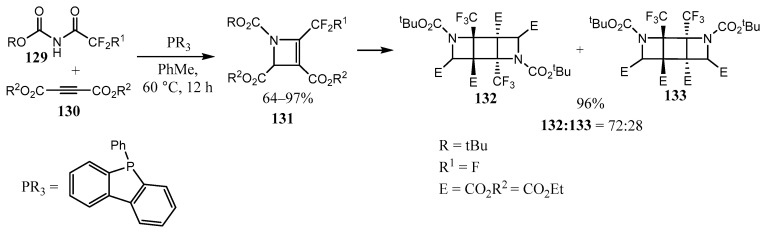
Phosphine-promoted synthesis of fluorinated 2-azetines **131** and their condensation to tricyclic diazetidines **132**–**133**.

**Figure 34 ijms-26-07435-f034:**
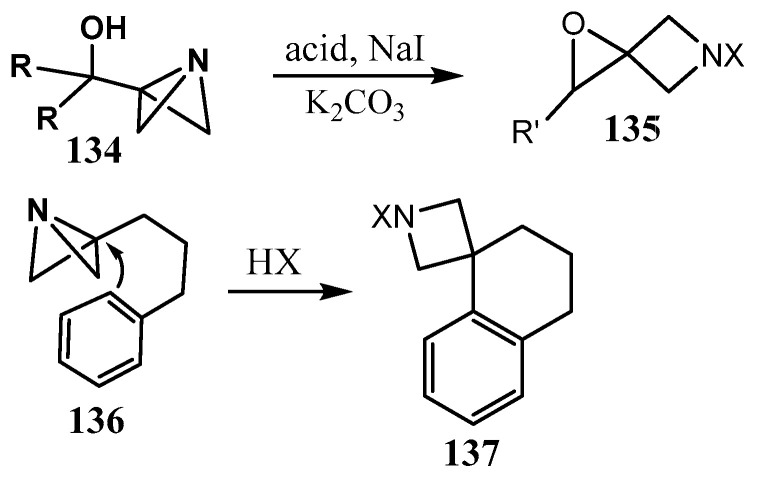
Azabicyclo[1.1.0]butane ring opening followed by intramolecular cyclization to the spiro-fused 2-azetidines.

**Figure 35 ijms-26-07435-f035:**
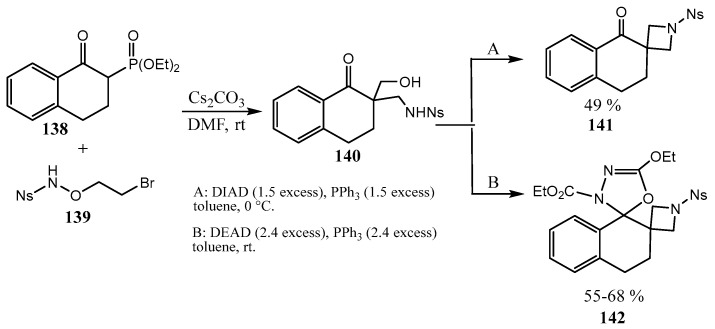
Synthesis of spirocyclic azetidines in the presence of DIAD (diisopropyl azodicarboxylate) or DEAD (diethyl azodicarboxylate).

**Figure 36 ijms-26-07435-f036:**
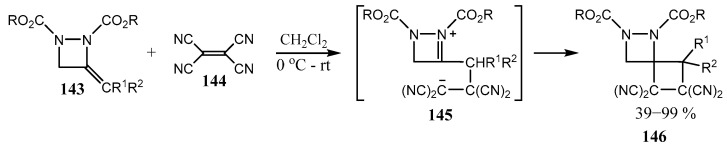
Synthesis of diazaspiro[3.3]heptanes **146** by reaction of tetracyanoethylene **144** with 3-alkylidene-1,2-diazetidines **143**.

**Figure 37 ijms-26-07435-f037:**
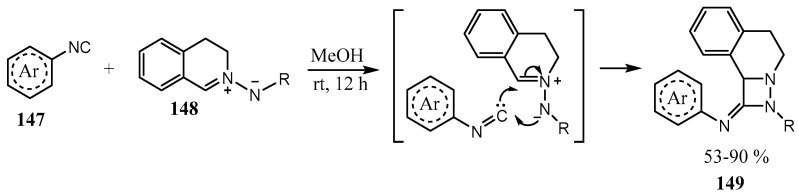
Synthesis of strained 1,2-diazetidines by [3 + 1] cycloaddition of isocyanides to azomethine imines.

**Figure 38 ijms-26-07435-f038:**
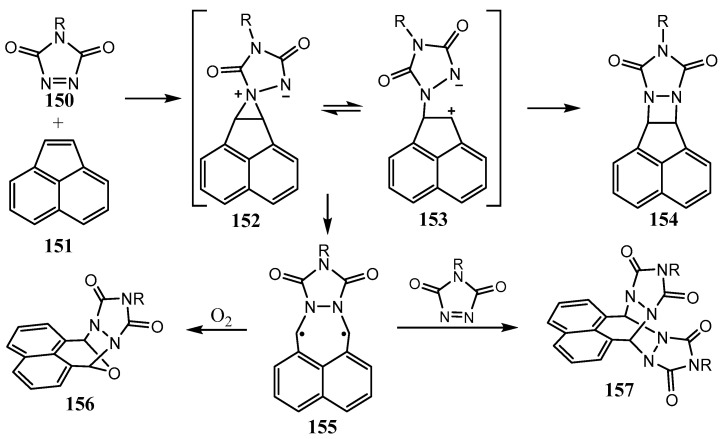
Reaction of *N*-methyl-1,2,4-triazoline-3,5-dione with acenaphthylene.

**Figure 39 ijms-26-07435-f039:**
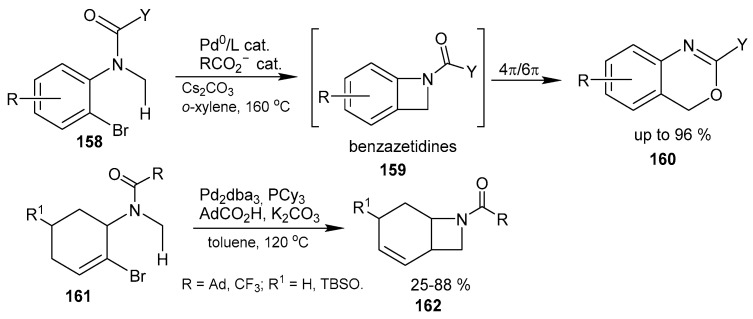
Syntheses proceeding via azetidines (**above**) and leading to azetidines (**below**).

**Figure 40 ijms-26-07435-f040:**
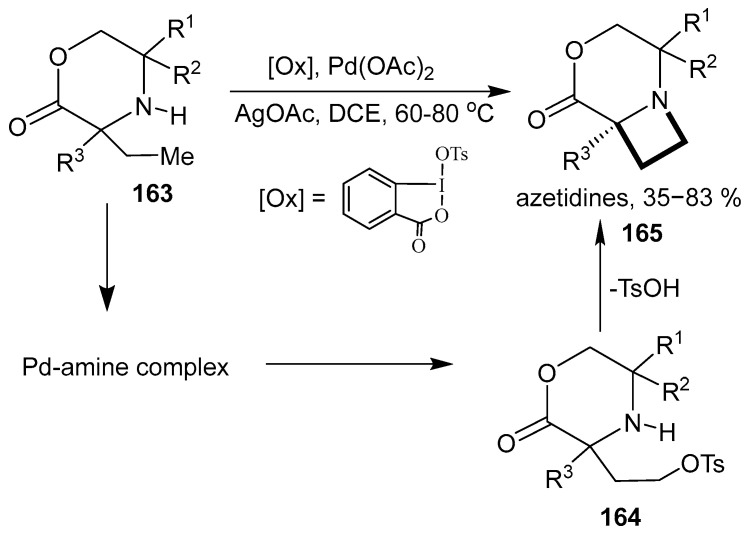
Synthesis of azetidines by the reaction of C–H amination.

**Figure 41 ijms-26-07435-f041:**
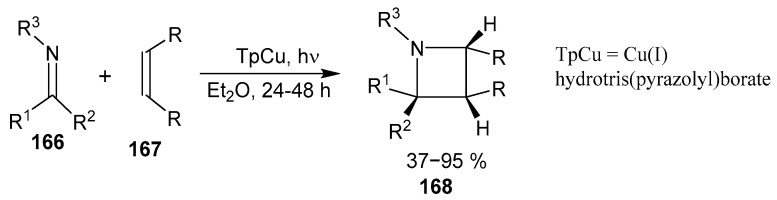
Azetidines via substrate coordination with the alkene π-component in a [2 + 2]- imine-olefin photocycloaddition.

**Figure 42 ijms-26-07435-f042:**
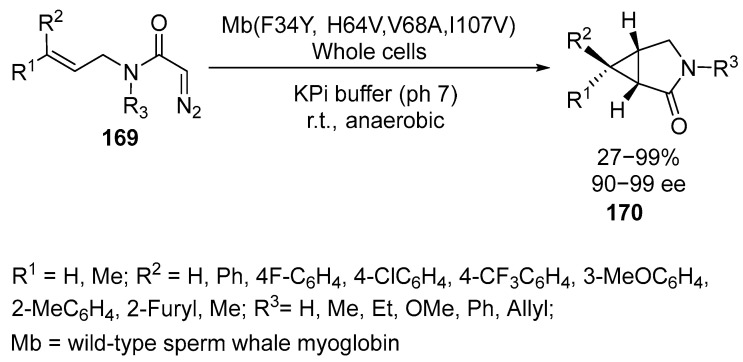
Intramolecular cyclopropanation of allyl-α-diazoacetamides **169** with Mb.

**Figure 43 ijms-26-07435-f043:**
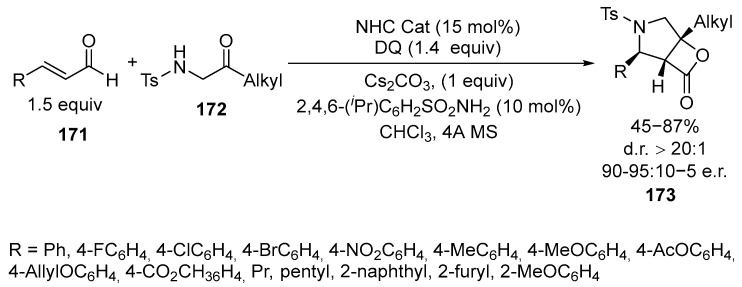
Asymmetric catalytic synthesis of bicyclic b-lactones **173** with a fused pyrrolidine ring from enals **171** and α-amino ketones **172**.

**Figure 44 ijms-26-07435-f044:**
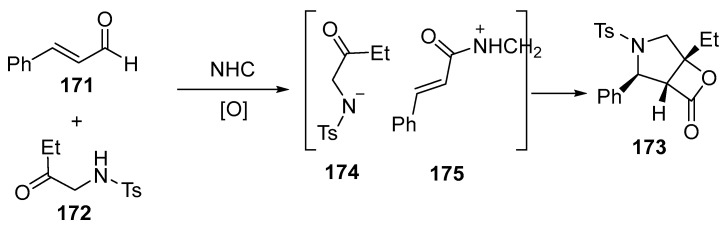
Proposed mechanism of the synthesis of a fused pyrrolidine ring from enals and α-amino ketones in the presence of NHC.

**Figure 45 ijms-26-07435-f045:**
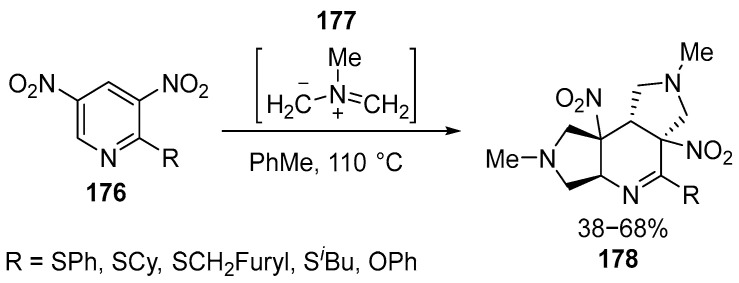
[3 + 2]-Cycloaddition of *N*-methyl azomethine ylide to 2-substituted 3,5-dinitropyridines.

**Figure 46 ijms-26-07435-f046:**
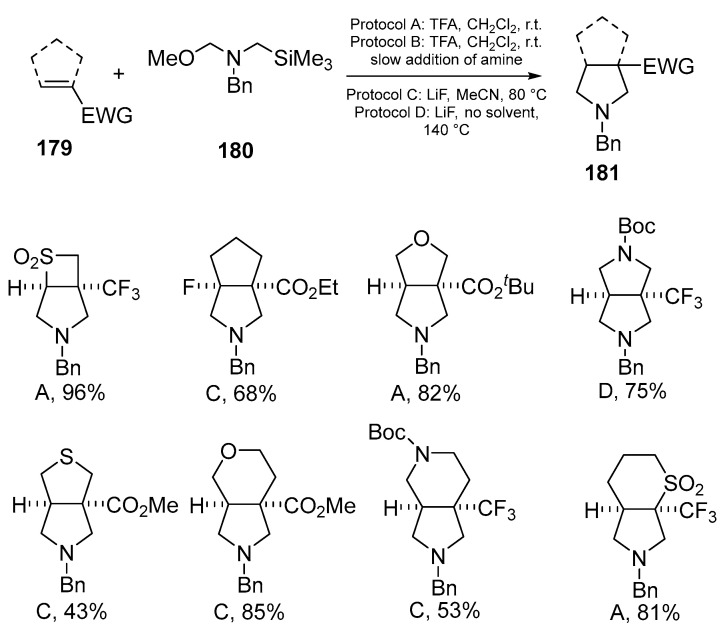
Synthesis of bicyclic pyrrolidines in the reaction [3 + 2]-cycloaddition.

**Figure 47 ijms-26-07435-f047:**
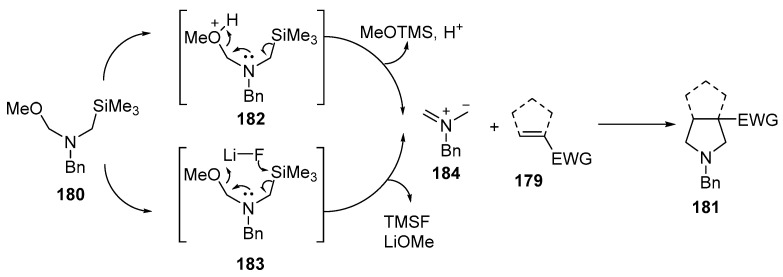
Mechanism of [3 + 2]-cycloaddition.

**Figure 48 ijms-26-07435-f048:**
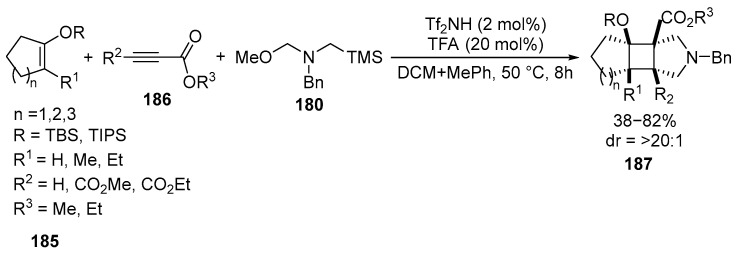
Synthesis of polysubstituted fused pyrrolidines **187** via [2 + 2]/[2 + 3] cycloaddition of azomethine ylides.

**Figure 49 ijms-26-07435-f049:**
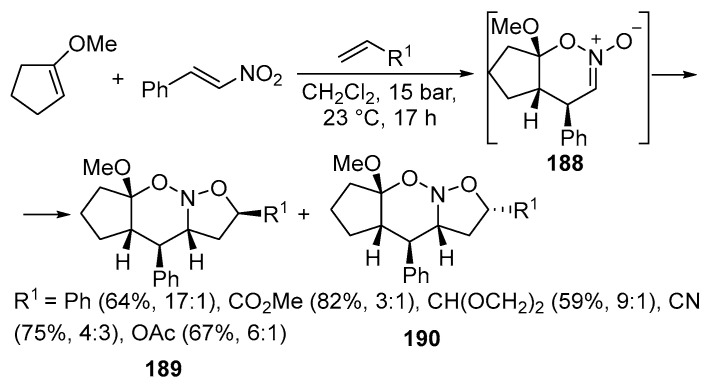
Multicomponent reactions of enol ether to form azonites **189**–**190**.

**Figure 50 ijms-26-07435-f050:**
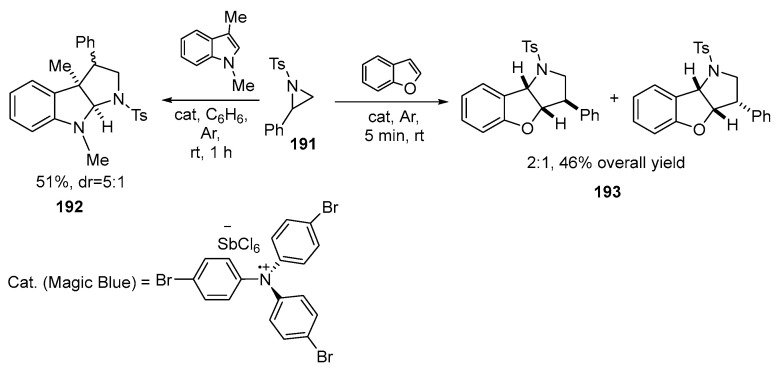
Magic Blue-initiated ring opening of non-racemic aziridine and DROC of aziridine **191**.

**Figure 51 ijms-26-07435-f051:**
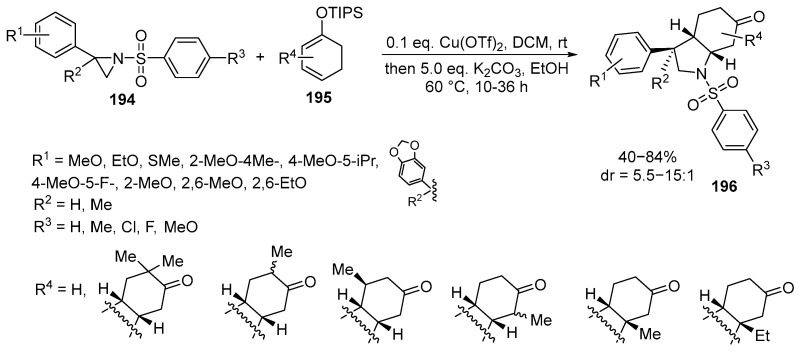
[3 + 2]-Cyclization of aziridines and silyl dienol ethers.

**Figure 52 ijms-26-07435-f052:**
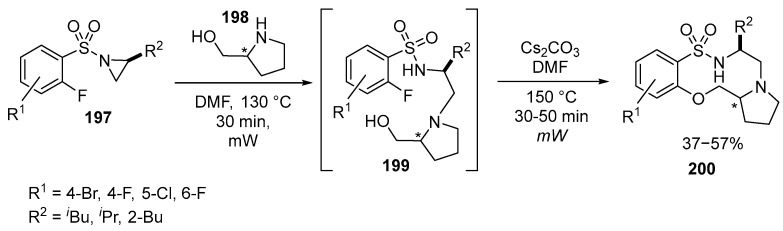
Synthesis of 10-membered benzo-fused sultams in one-pot, sequential aziridine ring opening with amino alcohols.

**Figure 53 ijms-26-07435-f053:**
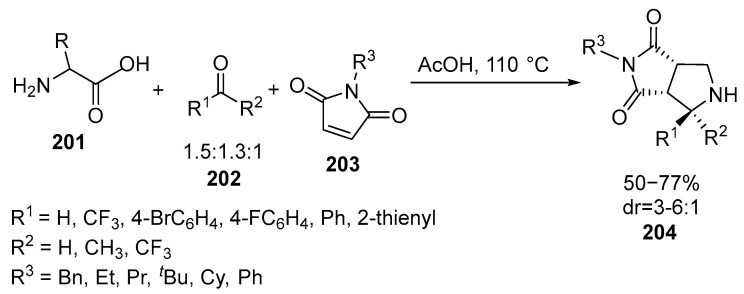
Trifluoromethylated fused pyrrolidines via decarboxylative [3 + 2]-cycloaddition of non-stabilized *N*-unsubstituted azomethine ylides.

**Figure 54 ijms-26-07435-f054:**
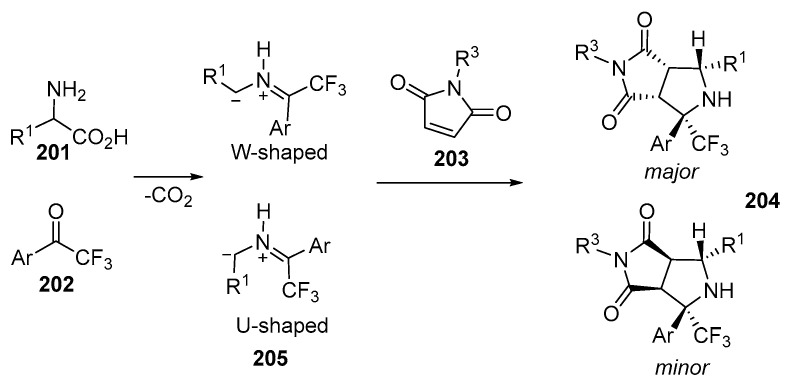
Stereochemistry in the cycloaddition of azomethine ylides to maleimides.

**Figure 55 ijms-26-07435-f055:**
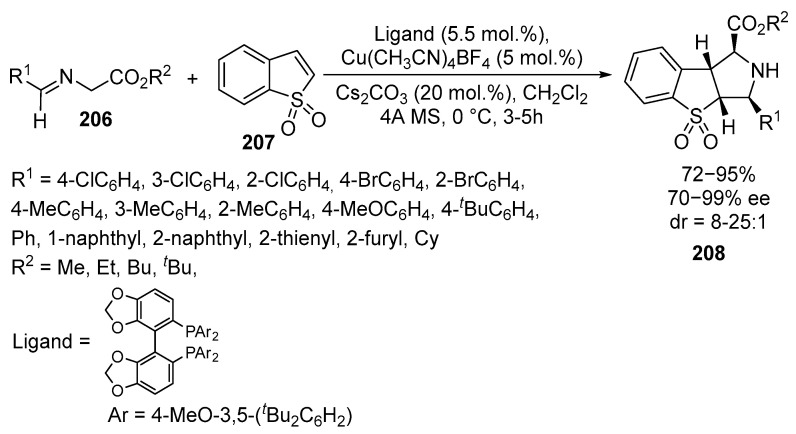
Enantioselective design of tricyclic pyrrolidine-fused benzo[b]thiophene 1,1-dioxide derivatives via copper(I)-catalyzed asymmetric 1,3-dipolar cycloaddition.

**Figure 56 ijms-26-07435-f056:**
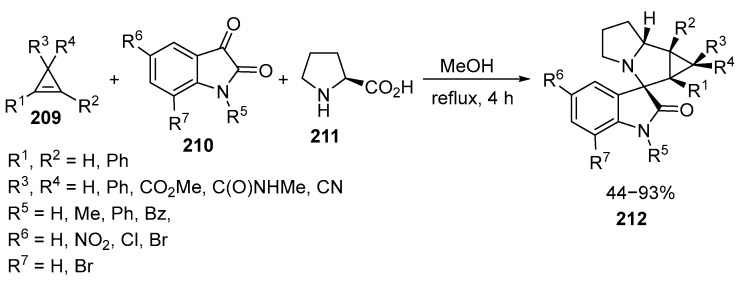
Synthesis of 3-spiro[cyclopropa[*a*]pyrrolizines] via one-pot three-component reactions of isatins, L-proline, and cyclopropenes.

**Figure 57 ijms-26-07435-f057:**
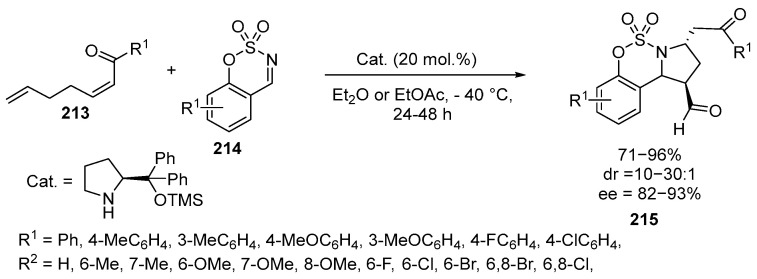
Catalytic enantioselective cycloaddition of cyclic N-sulfimines.

**Figure 58 ijms-26-07435-f058:**
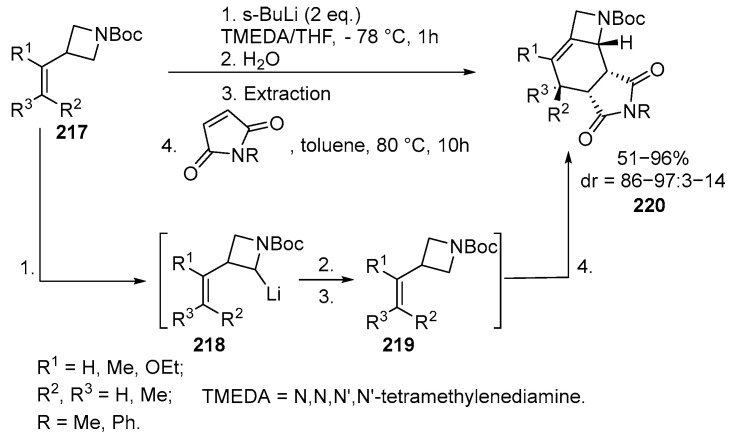
2- and 3-Alkylideneazetines in the reaction with substituted maleimides.

**Figure 59 ijms-26-07435-f059:**
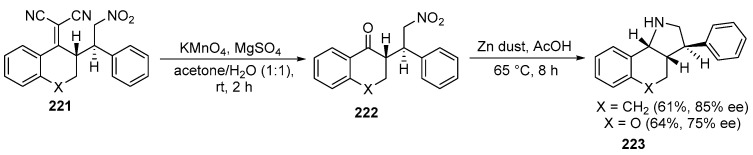
Synthesis of polycyclic fused pyrrolidines.

**Figure 60 ijms-26-07435-f060:**
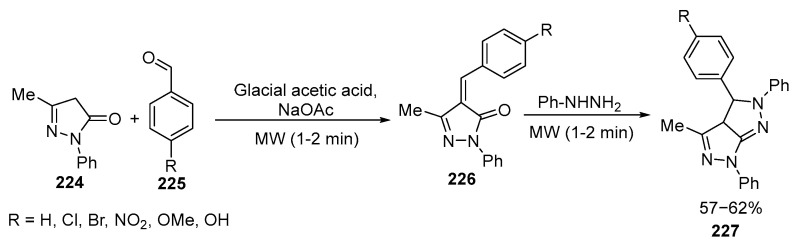
Synthesis of substituted **227**.

**Figure 61 ijms-26-07435-f061:**
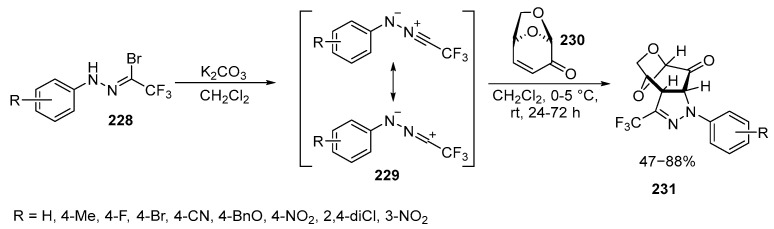
(3 + 2)-Cycloadditions of levoglucosenone with fluorinated nitrile imine.

**Figure 62 ijms-26-07435-f062:**
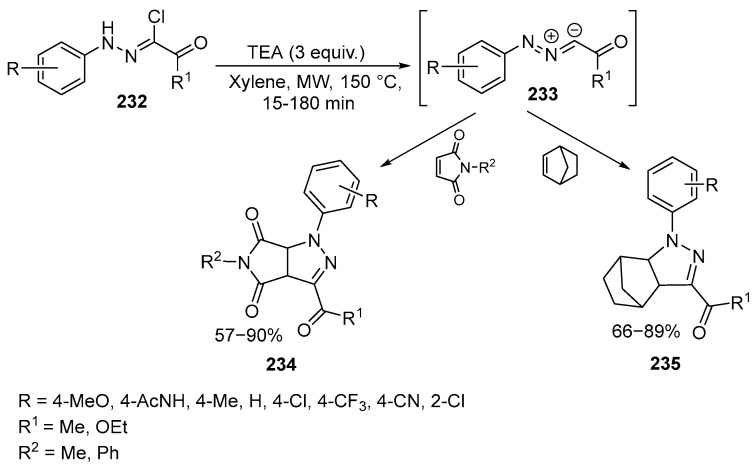
Cycloaddition of hydrazonoyl chlorides with dipolarophiles.

**Figure 63 ijms-26-07435-f063:**
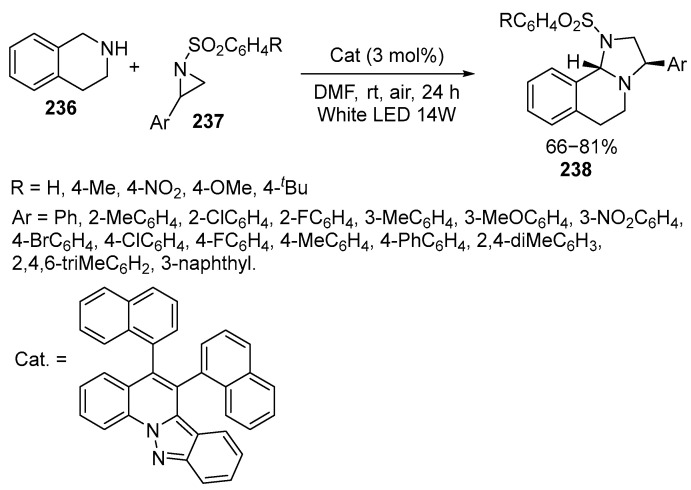
Assembling of fused imidazolidines via tandem ring opening/oxidative amination of aziridines with cyclic secondary amines using photoredox catalysis.

**Figure 64 ijms-26-07435-f064:**
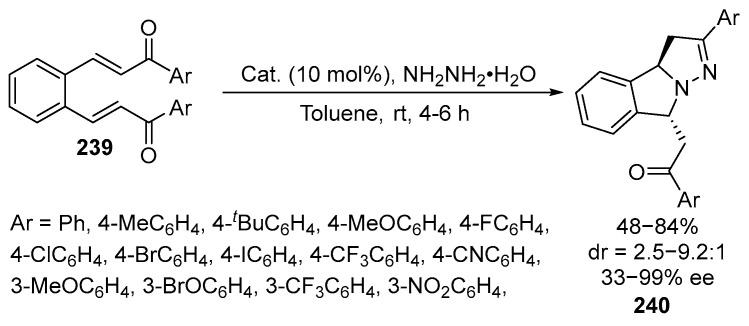
Catalytic asymmetric free hydrazine addition to synthesize chiral fused pyrazolines.

**Figure 65 ijms-26-07435-f065:**
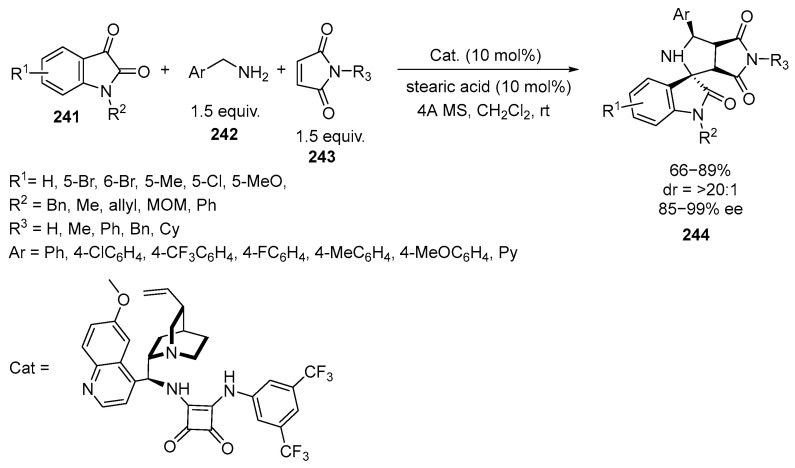
Synthesis of chiral pyrrolidine-fused spirooxindoles via organocatalytic [3 + 2] 1,3-dipolar cycloaddition of azomethine ylides with maleimides.

**Figure 66 ijms-26-07435-f066:**
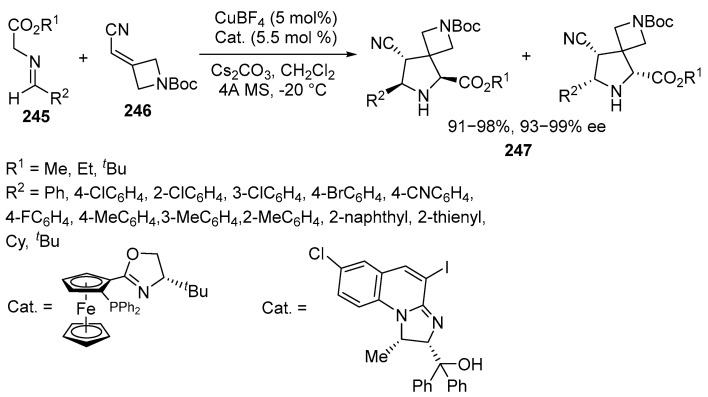
Catalytic asymmetric construction of spirocyclic pyrrolidine-azetidine.

**Figure 67 ijms-26-07435-f067:**
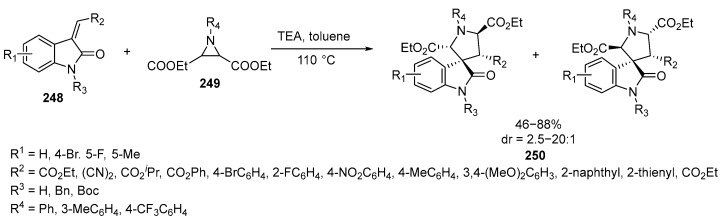
Synthesis of fully substituted pyrrolidine-fused 3-spirooxindoles via 1,3-dipolar cycloaddition of aziridine and 3-ylideneoxindole.

**Figure 68 ijms-26-07435-f068:**
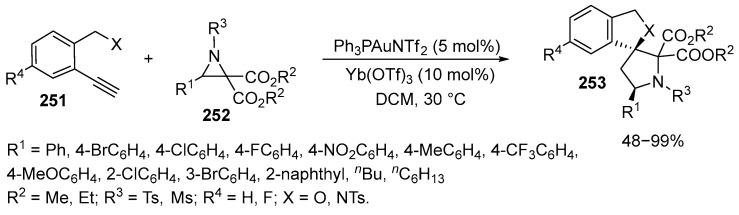
Au-catalyzed cycloisomerization/diastereoselective [3 + 2]-cycloaddition.

**Figure 69 ijms-26-07435-f069:**
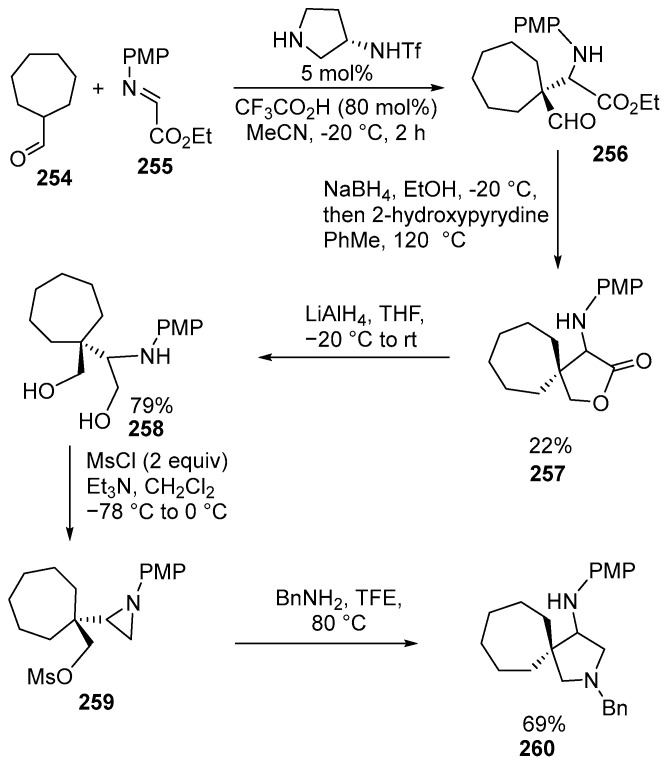
Organocatalytic assembling of spiro[4.6]undecanes containing 3-aminopyrrolidines.

**Figure 70 ijms-26-07435-f070:**
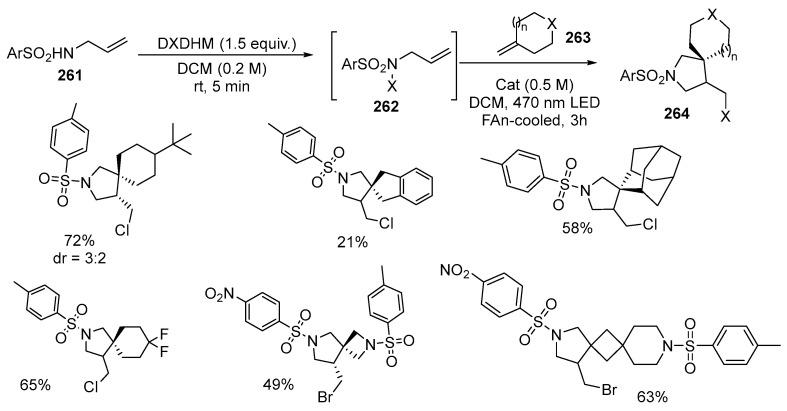
Assembling of *β*-spirocyclic pyrrolidines from *N*-allylsulfonamides and alkenes.

**Figure 71 ijms-26-07435-f071:**
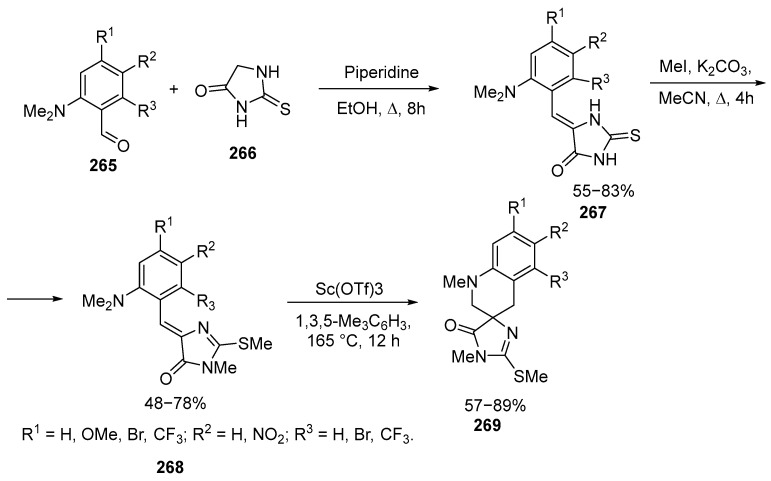
Synthesis of spiro[imidazole-4,3′-quinolin]ones.

**Figure 72 ijms-26-07435-f072:**
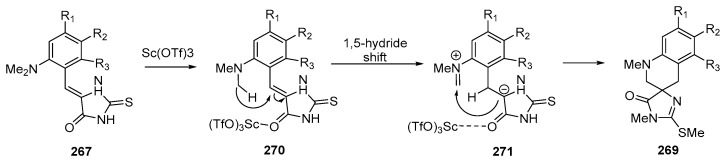
Proposed mechanism of the formation of spiro[imidazole-4,3′-quinolin]ones.

**Figure 73 ijms-26-07435-f073:**
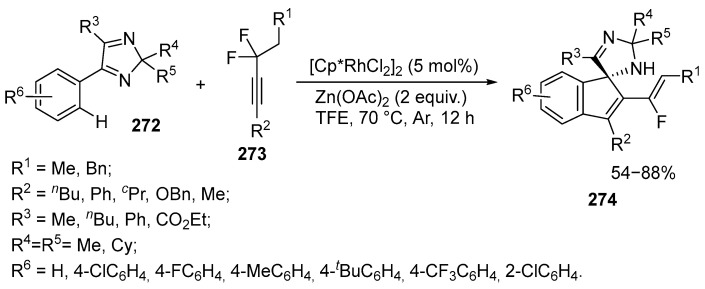
Synthesis of fluorovinyl spiro-[imidazole-indene] in the presence of Rh(III)-catalyst.

**Figure 74 ijms-26-07435-f074:**
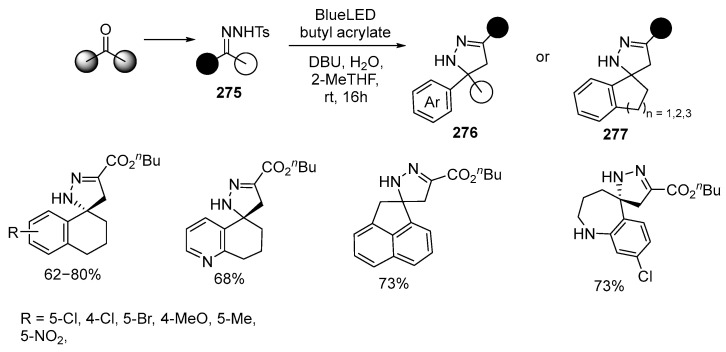
Blue-LED [3 + 2] cycloadditions of donor/donor diazo intermediates with alkenes achieve (spiro)-pyrazolines **276** or **277**.

**Figure 75 ijms-26-07435-f075:**
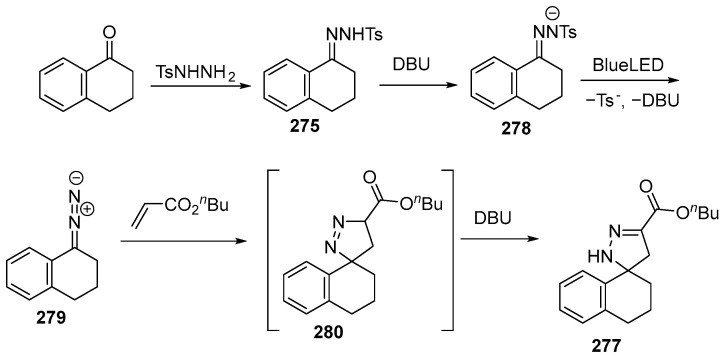
Proposed mechanism of Blue-LED [3 + 2] cycloadditions of donor/donor diazo intermediates with alkenes to achieve (spiro)-pyrazolines.

**Figure 76 ijms-26-07435-f076:**
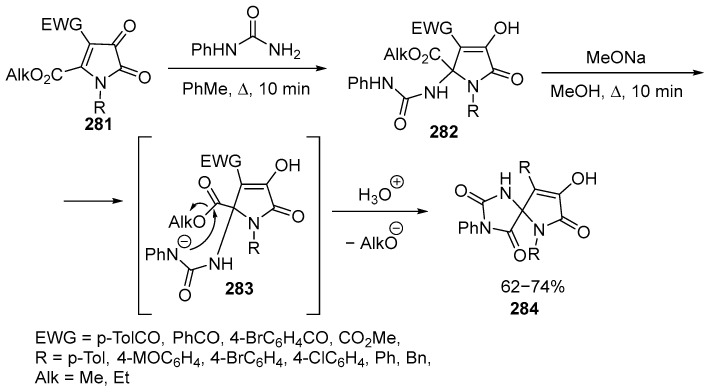
Imidazole spiro compounds from 5-alkoxycarbonyl to 1H-pyrrole-2,3-diones and phenylurea.

**Figure 77 ijms-26-07435-f077:**
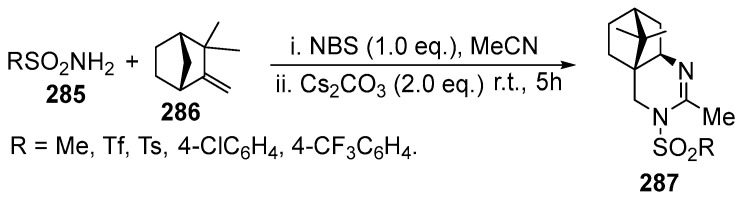
Heterocyclization of sulfonamides RSO_2_NH_2_ with camphene.

**Figure 78 ijms-26-07435-f078:**
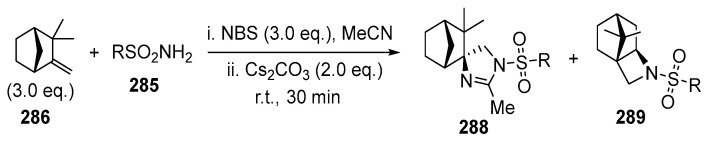
Heterocyclization of camphene (3 eq.) with sulfonamides in the presence of NBS (3 eq.) and Cs_2_CO_3_ (2 eq.) in MeCN.

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
