# Peer review of "Fused-Linked and Spiro-Linked N-Containing Heterocycles"

_ijms, 2025, doi:10.3390/ijms26157435_

Round 1

Reviewer 1 Report

Comments and Suggestions for Authors

The review by M. Yu. Moskalik and B. A. Shainyan is devoted to azaheterocycles, including different sizes of rings and their combination into polycyclic systems. Interesting and large material devoted to synthetic chemistry is combined, including various methods, including cycloaddition and metal-catalyzed reactions.

Some comments on the text.

  1. Scheme 4. Check the correctness of the last compound. In the structure of the product, the carbon atom with the carbonyl group is surprising.
  2. The numbering of the compounds is missing, which somewhat complicates the text itself for perception. I would recommend introducing numbering and supplementing the text with their mention.
  3. In continuation of comment number two. The full names of the compounds in the text can be replaced if you introduce numbering. Since the full names of the compounds are often used for the experimental part, then for the review this only complicates the narrative.
  4. Scheme 15. And throughout the text it is necessary to correct the index at R substituents and move it to upper case.
  5. Scheme 24 In the initial compound, nitrogen is divalent, perhaps hydrogen is missing?
  6. After Scheme 33, Scheme 1 comes again!
  7. In the explanation of Scheme 73, 74, explain the role of Sc(III) salt. Scheme 75, correct the NH group in heterocycles.

Author Response

Dear Reviewer,

Please find attached the revised manuscript with a title “Fused-linked and Spiro-linked N-Containing Heterocycles” (ijms-3751221) in which we have taken considerable effort to respond to the Editor`s and Reviewers` suggestions and comments. We thank you for the opportunity to resubmit our manuscript.

We appreciate the time and efforts spent by Editor and Reviewers to critically review our manuscript. We made the corrections and revised the text according to the comments.

We hope now it will be acceptable for publication in International Journal of Molecular Sciences.

Below the step-by-step answers to all comments are given.

Reviewer 1.

  1. Scheme 4. Check the correctness of the last compound. In the structure of the product, the carbon atom with the carbonyl group is surprising.

We are sorry, the structure was wrong. Now corrected.

  1. The numbering of the compounds is missing, which somewhat complicates the text itself for perception. I would recommend introducing numbering and supplementing the text with their mention.

Done. We introduced numbering and supplementing the text with their mention.

  1. In continuation of comment number two. The full names of the compounds in the text can be replaced if you introduce numbering. Since the full names of the compounds are often used for the experimental part, then for the review this only complicates the narrative.

Done. Most of the full names of the compounds have been changed to numbers.

  1. Scheme 15. And throughout the text it is necessary to correct the index at R substituents and move it to upper case.

Done.

  1. Scheme 24 In the initial compound, nitrogen is divalent, perhaps hydrogen is missing?

We are sorry for this misprint. Now corrected.

  1. After Scheme 33, Scheme 1 comes again!

We are sorry for this misprint. Now corrected as Scheme 34.

  1. In the explanation of Scheme 73, 74, explain the role of Sc(III) salt. Scheme 75, correct the NH group in heterocycles.

The paragraph above Figure 73 is now revised. The presentation of NH groups in heterocycles in Scheme 75 are unified.

Reviewer 2 Report

Comments and Suggestions for Authors

Fused-linked and Spiro-linked N-Containing Heterocycles

Mikhail Yu. Moskalik and Bagrat A. Shainyan

The article is devoted to an important class of fused and spiro-nitrogen-containing heterocyclic compounds. Writing such review articles helps researchers better understand the large number of publications on this topic. Therefore, I believe that this review is appropriate for this journal.

There are a number of comments.

  1. An abstract should be drawn graphically to make it easier to understand what the review article is about.
  2. The Introduction should describe the principle by which the authors divide the article into chapters. There are a huge number of articles on this topic, the authors should provide time boundaries for the cited literature.
  3. Several review articles on this topic should be cited in the Introduction so that readers can understand the difference between the reviews.
  4. IJMS is an interdisciplinary journal. The authors should indicate what biological activity the obtained compounds have in Figure 1 through Figure 79. I understand that not all compounds have biological activity, but we should try to do this.
  5. The authors write Scheme everywhere in the text. For example, "Both reactions are [2+1]-cycloaddition processes (Scheme 1)." However, under the schemes it is written Figure. For example, "Figure 1. Two alternative routes to fused aziridines." Everywhere "Figure" should be replaced with "Scheme".
  6. In all reaction schemes it is necessary to write the yield of the product. For example, Schemes 2,3.
  7. Figure 4. As I understand it, a lot of compounds are obtained, because it is written "Ar" and "Ar'". At the same time, the yield is indicated as 98 and 61%. It is necessary to indicate the interval. Sometimes the yield of the product is indicated under the arrow, sometimes under the compound. It is necessary to do it uniformly.
  8. Figure 5. In the scheme, write the structure of the Cu(II) complex. 9. Figure 6. The final compound has several chiral centers. Indicate the enantiomeric or diastereomeric excess in the diagram. This applies to all diagrams that yield products with chiral centers. This is very important, since even enantiomers can exhibit different biological activity. The text says aza-Darzens reaction, while the diagram shows the Mannich reaction.
  9. Figure 13. Explain the structure of ArSO2NNaBr. In this diagram and others, the reaction conditions (solvent, temperature, and time) should be described in more detail.
  10. Abbreviations should be explained in the text or diagram the first time they are mentioned. For example, Ts, Tol, tBuS(O),TBAF, m-CPBA, DIAD, Boc, TMEDA, DABCO, TCNE, TMSF, DBU, TMS.
  11. Figure 20. The paragraph related to Scheme 20 contains text to Scheme 19. The text to Scheme 19 should be moved above to Scheme 19.
  12. Figure 18. "The authors of [50] disclosed an efficient diastereoselective synthesis of N-alkyl spi- 214 roaziridines by addition of primary amines to α,β-unsaturated ketones in the oxidative 215 system I2/tBuOOH, similar to the reaction in Scheme 15." It is not entirely clear which α,β-unsaturated ketones are being discussed. Scheme 18 is unclear. Where is the Ts group in the intermediate in square brackets? Is there a TolSO fragment or Ts in the original compound?
  13. Figure 21. References [47] and [48] are indicated under the arrows in the scheme. Is that correct? These references are not mentioned in the text.
  14. Throughout the text, it is necessary to write in italics: dr, N-unsubstituted, (R)- and (S)-prolinol, ee, s-Bu.
  15. "Note that the work [67], in which (DHQD)2PHAL, hydroquinidine 1,4-phthalazi-nediyl diether, was used as a chiral catalyst, was the first enantioselective Neber reaction of O-sulfonyl ketoxime, and allowed to synthesize spirocyclic oxindoles with the azirine motif in good to excellent yields and with up to a 92:8 enantiomeric ratio." This paragraph refers to Scheme 21. It should also be moved above, where these reactions are discussed. This is a general comment. The authors should correct it throughout the text.
  16. The text does not reference "Figure 21." This paragraph only references Scheme 19.
  17. Figure 23. The yield of 85-89% of the product is unclear. One product is obtained. Or does it depend on the alcohol? A comment should be given on this issue.
  18. Figure 26. The authors write "chiral N-heterocyclic carbene" and "chiral phosphoric acid". The choice of catalyst is important for enantioselective reactions. Therefore, it is necessary to write which catalysts were used.
  19. Figure 28. It is necessary to write what kind of diastereoselectivity.
  20. Figure 33. It is necessary to check the correctness of the structure isolated with a yield of 64-97%. And with what yield were the other products isolated? Were they isolated at all or were they formed as a mixture? Where did the substituents E and tBuO2C come from? It is necessary to write the reaction conditions in more detail. Is primary phosphine used in the first stage? Which one?
  21. "Figure 1. azabicyclo[1.1.0]butane ring followed by intramolecular cyclization to the spiro-fused azetidines .". It should be replaced with "Figure 34".
  22. Figure 38. The structure of the second compound is unclear? How stable is this betaine?
  23. Figure 42. [2+2] cycloaddition reactions are stereoselective. What is the stereoselectivity of this reaction? This should be discussed in the text and shown in the diagram.
  24. Figure 47. The diagram shows Protocol A, but Protocol B is missing.
  25. Figure 74. R1, R2, R3 should be written as numbers in subscript.
  26. Figure 76. Everywhere the designated hydrogen should be written in italics. For example, here "tetrahydro-4H-indol-4-one and 7,8-dihydroquinolin-5(6H)-one". "N-tosylhydrazones" N- should be written in italics.
  27. Figure 78 and Figure 79. These diagrams show similar reactions: substrates and conditions. Why are the products different?
  28. Conclusions. The authors listed the reactions in the conclusions. It is necessary to give a personal assessment, it is necessary to note promising directions, dead-end directions, which direction should be developed, etc.

Only after all comments have been corrected can the article be accepted into the journal.

Author Response

Dear Reviewer,

Please find attached the revised manuscript with a title “Fused-linked and Spiro-linked N-Containing Heterocycles” (ijms-3751221) in which we have taken considerable effort to respond to the Editor`s and Reviewers` suggestions and comments. We thank you for the opportunity to resubmit our manuscript.

We appreciate the time and efforts spent by Editor and Reviewers to critically review our manuscript. We made the corrections and revised the text according to the comments.

We hope now it will be acceptable for publication in International Journal of Molecular Sciences.

Below the step-by-step answers to all comments are given.

Reviewer 2.

There are a number of comments.

  1. An abstract should be drawn graphically to make it easier to understand what the review article is about.

Done

  1. The Introduction should describe the principle by which the authors divide the article into chapters. There are a huge number of articles on this topic, the authors should provide time boundaries for the cited literature.

Done. Information was added in the Introduction.

  1. Several review articles on this topic should be cited in the Introduction so that readers can understand the difference between the reviews.

Done. Information was added in the Introduction.

  1. IJMS is an interdisciplinary journal. The authors should indicate what biological activity the obtained compounds have in Figure 1 through Figure 79. I understand that not all compounds have biological activity, but we should try to do this.

The authors endeavored to gather as much information as possible about the biological effects of the substances in question, for example, the data provided in Lines 96, 99, 219, 265, 506, 517, 573, 674, 690, 720 and others.

  1. The authors write Scheme, however, under the schemes it is written Figure.

The Reviewer is right, now the whole text is revised according to the format of the Journal.

  1. In all reaction schemes it is necessary to write the yield of the product. For example, Schemes 2,3.

The interval is now given for the final products. Its precursors are formed almost quantitatively.

  1. Figure 4. The yield is indicated as 98 and 61%. It is necessary to indicate the interval.

Done.

  1. Figure 5. In the scheme, write the structure of the Cu(II) complex.

Done. Information was added

  1. Figure 6. The final compound has several chiral centers. Indicate the enantiomeric or diastereomeric excess in the diagram. This applies to all diagrams that yield products with chiral centers. This is very important, since even enantiomers can exhibit different biological activity. The text says aza-Darzens reaction, while the diagram shows the Mannich reaction.

Done. We are sorry for this misprint. Now corrected.

  1. Figure 13. Explain the structure of ArSO2NNaBr. In this diagram and others, the reaction conditions (solvent, temperature, and time) should be described in more detail.

Done. Information was added

  1. Abbreviations should be explained. Ts, Tol, tBuS(O),TBAF, m-CPBA, DIAD, Boc, TMEDA, etc.

Most are explained, but Ts, Tol, and some other are well known and do not need explanation.

  1. Figure 20. The paragraph related to Scheme 20 contains text to Scheme 19. The text to Scheme 19 should be moved above to Scheme 19.

Done. We are sorry for this misprint. Now corrected.

  1. Figure 18. "The authors of [50] disclosed an efficient diastereoselective synthesis of N-alkyl spiroaziridines by addition of primary amines to α,β-unsaturated ketones in the oxidative system I2/tBuOOH, similar to the reaction in Scheme 15." It is not entirely clear which α,β-unsaturated ketones are being discussed. Scheme 18 is unclear. Where is the Ts group in the intermediate in square brackets? Is there a TolSO fragment or Ts in the original compound?

Scheme was corrected. We are sorry for this misprint. Now corrected.

  1. Figure 21. References [47] and [48] are indicated under the arrows in the scheme. Is that correct? These references are not mentioned in the text.

Scheme was corrected. We are sorry for this misprint. Now corrected.

  1. Throughout the text, it is necessary to write in italics: dr, N-unsubstituted, (R)-, (S)-prolinol, ee, s-Bu.

Done.

  1. "Note that the work [67], in which (DHQD)2PHAL, hydroquinidine 1,4-phthalazi-nediyl diether, was used as a chiral catalyst, was the first enantioselective Neber reaction of O-sulfonyl ketoxime, and allowed to synthesize spirocyclic oxindoles with the azirine motif in good to excellent yields and with up to a 92:8 enantiomeric ratio." This paragraph refers to Scheme 21. It should also be moved above, where these reactions are discussed. This is a general comment. The authors should correct it throughout the text.

Done.

  1. The text does not reference "Figure 21." This paragraph only references Scheme 19.

Done.

  1. Figure 23. The yield of 85-89% is unclear. ... Or does it depend on the alcohol?

The Reviewer is right, the yield depends on the alcohol, which, in our opinion, is evident.

  1. 26. "chiral NHC" and "chiral phosphoric acid". it is necessary to write which catalysts were used.

Done.

  1. Figure 28. It is necessary to write what kind of diastereoselectivity.

In our opinion, it is clear: the content of diastereomer with cis-arranged R1 and R4 is 99%.

  1. Figure 33. It is necessary to check the correctness of the structure isolated with a yield of 64-97%. And with what yield were the other products isolated? Were they isolated at all or were they formed as a mixture? Where did the substituents E and tBuO2C come from? It is necessary to write the reaction conditions in more detail. Is primary phosphine used in the first stage? Which one?

The structure is correct. The yield of other products – 96% (mixture). The second part of Figure 33 – is the example for one concrete substrate, where R1 = F, R2 = tBu, E =R1 = CO2Et.

Scheme was corrected. We are sorry for this misprint. Now corrected.

  1. "Figure 1. should be replaced with "Figure 34".

Done.

  1. Figure 38. The structure of the second compound is unclear? How stable is this betaine?

We think that everything is clear, but decided to detalize the scheme to make it even more clear.

  1. Figure 42. [2+2] cycloaddition is stereoselective. This should be shown in the diagram.

Now it is shown in the scheme. To discuss in the text in all such cases would take too much place.

  1. Figure 47. The diagram shows Protocol A, but Protocol B is missing.

Corrected. Protocol B is included in Figure 47.

  1. Figure 74. R1, R2, R3 should be written as numbers in subscript.

Done.

  1. Figure 76. Everywhere the designated hydrogen should be written in italics.

Done.

  1. Figure 78 and 79 show similar reactions: substrates and conditions. Why are the products different?

Corrected. A small paragraph is now added to explain the difference in the reaction courses.

  1. The authors listed the reactions in the conclusions. It is necessary to give a personal assessment, to note promising and dead-end directions, which direction should be developed, etc.

The conclusion has been corrected, some information has been added.

Reviewer 3 Report

Comments and Suggestions for Authors

This review article provides a detailed exploration of synthetic methods for creating fused and spiro nitrogen-containing heterocycles, with a focus on three- to six-membered rings such as aziridines, azetidines, pyrrolidines, pyrazolines, and imidazolines. The review encompasses a wide array of modern techniques, including cycloadditions, ring closures, and ring expansions catalyzed by metals, photochemical, or organocatalytic methods. The importance of these structures in pharmaceutical and materials chemistry is clearly highlighted. The review cites over 170 papers, offering a broad and current overview of the field. Most references are from 2014 to 2024, reflecting the authors' focus on recent advancements. Reaction schemes and figures elucidate complex pathways and mechanisms. The authors often comment on selectivity, rearrangements, and unexpected outcomes, adding value to the data. This balanced approach emphasizes recent methodologies while maintaining a historical perspective where relevant.

I find this review informative and recommend it for publication in IJMS. However, a few enhancements could improve its clarity and utility:

  • Consider adding a comparative table summarizing synthetic methods, including ring size, reagents, conditions, yields, and selectivity. This would make the information more accessible and easier to apply.
  • While the article touches on biological relevance, it lacks concrete examples or data points, which could make the content more application-oriented. I propose to expand the section on biological activity to include experimental data, pharmacological assays, or specific therapeutic targets.
  • For intricate reactions like rearrangements and cycles expansions, more illustrations or step to step explanations would be beneficial. Please provide explicit mechanistic schemes for non-intuitive transformations, such as pseudo-sigmatropic rearrangements or ring expansions.
  • The article does not address synthetic issues such as regioselectivity, scalability, or handling unstable intermediats. What are the primary synthetic or practical limitations in applying these methods on a preparative scale?

Author Response

Dear Reviewer,

Please find attached the revised manuscript with a title “Fused-linked and Spiro-linked N-Containing Heterocycles” (ijms-3751221) in which we have taken considerable effort to respond to the Editor`s and Reviewers` suggestions and comments. We thank you for the opportunity to resubmit our manuscript.

We appreciate the time and efforts spent by Editor and Reviewers to critically review our manuscript. We made the corrections and revised the text according to the comments.

We hope now it will be acceptable for publication in International Journal of Molecular Sciences.

Below the step-by-step answers to all comments are given.

Reviewer 3.

Few enhancements could improve its clarity and utility:

  • Consider adding a comparative table summarizing synthetic methods, including ring size, reagents, conditions, yields, and selectivity. This would make the information more accessible and easier to apply.

The authors would like to thank the reviewer for their suggestion. However, given the wide range of chemical transformations discussed in the review, it was necessary to include detailed information on most of the proposed reaction schemes in this table. While this may not make the information easier to access or apply,

  • While the article touches on biological relevance, it lacks concrete examples or data points, which could make the content more application-oriented. I propose to expand the section on biological activity to include experimental data, pharmacological assays, or specific therapeutic targets.

Since a detailed discussion of biological activity-related issues is beyond the scope of this review and the scope of authors` competence, it seems that the authors would have difficulty in devoting a separate section to this topic. Nevertheless, Authors have included data on various forms of biological activity that were found in the papers reviewed.

  • For intricate reactions like rearrangements and cycles expansions, more illustrations or step to step explanations would be beneficial. Please provide explicit mechanistic schemes for non-intuitive transformations, such as pseudo-sigmatropic rearrangements or ring expansions.

Done. Information was added. Details about pseudo-sigmatropic rearrangements and ring expansions was added in the Figure 11.

  • The article does not address synthetic issues such as regioselectivity, scalability, or handling unstable intermediats.

The issues of regioselectivity or handling unstable intermediats are widely discussed in this review. In those papers where the possibility of scalability is discussed, the authors also provided information on the review pages (e.g. Line 96, Line 592, Line 891)

What are the primary synthetic or practical limitations in applying these methods on a preparative scale?

Due to the wide range of methods considered in the review, it is very difficult to identify general limitations associated with the preparative applicability of the methods under consideration. However, as in general organic synthesis and medicinal chemistry, the availability of reagents, availability and cheapness of catalysts, and the use of anhydrous media impose their own limitations. It is worth noting that in the methods under consideration there are practically no examples of using high pressures or working with gaseous reagents.

Round 2

Reviewer 2 Report

Comments and Suggestions for Authors

-